# Efficiency-optimized relativistic plasma harmonics for extreme fields

Robin J. L. Timmis[1,2,10 ✉], Colm R. J. Fitzpatrick[3,10], Jonathan P. Kennedy[3,10], Holly M. Huddleston[3,10], Elliott Denis[1,10], Abigail James[1], Chris Baird[4], Dan Symes[4], David McGonegle[5], Eduard Atonga[1], Heath Martin[1], Jeremy Rebenstock[6], John Neely[5], Jordan Lee[1], Joshua Redfern[1], Nicolas Bourgeois[4], Oliver Finlay[4], Rusko Ruskov[1], Sam Astbury[4], Steve Hawkes[4], Zixin Zhang[1], Matt Zepf[7,8,9], Karl Krushelnick[6], Edward Gumbrell[5], Paramel Pattathil Rajeev[4], Mark Yeung[3,10], Brendan Dromey[3,10 ✉] & Peter Norreys[1,2,10]

Bright harmonic radiation from relativistically oscillating laser plasmas offers a direct route for generating extreme electromagnetic fields. Theory predicts that under optimized conditions, the plasma medium can support strong spatiotemporal compression of laser energy in a coherent harmonic focus (CHF), delivering intensity boosts many orders of magnitude greater than the incident driving laser pulse[1–4]. Although diffraction-limited performance[5] (spatial compression) and attosecond phase locking[6–8] (temporal compression) have been demonstrated experimentally, efficient coupling of relativistically intense laser pulse energy into the emitted harmonic cone has not been realized so far. Here we demonstrate that this highly nonlinear interaction can be tailored to deliver the maximum conversion efficiencies predicted from simulations. By fine-tuning the temporal profile of the driving laser on sub-picosecond ($<10^{-12}$ s) timescales, energies >9 mJ between the 12th and 47th harmonics are observed. These results are in agreement with the theoretically expected efficiency dependence on harmonic order, verifying that optimal conditions have been achieved in the generation process. This is the important final element required to achieve the expected intensity boosts from a CHF in experiments. Although obtaining spatiotemporal compression and optimal efficiency simultaneously remains challenging, the path to realizing extreme optical field strengths approaching the critical field of quantum electrodynamics (the Schwinger limit at $>10^{16}$ V cm$^{-1}$ or $>10^{29}$ W cm$^{-2}$) is now open, permitting all-optical studies of the quantum vacuum and new frontiers for intense attosecond science.

The generation of coherent extreme ultraviolet (XUV) and X-ray photons by high harmonic generation from solid targets (SHHG) relies on the formation of a steep electron density gradient tuned by the leading edge of a relativistically intense laser pulse interacting with a bulk, solid-density target. If this gradient is sufficiently short compared with the incident laser wavelength ($\lambda_L$), then the electromagnetic field can efficiently couple to this surface, driving electrons to oscillate coherently at near-light speeds, permitting strong frequency up-shifts in the reflected beam[9–14].

Along with this constraint, three core ingredients are required to achieve a CHF from relativistically oscillating plasmas and unlock the intensity boosts discussed above. First, the shorter wavelengths $\lambda_n = \lambda_L/n$ of higher harmonic orders, $n$, allow for the focusing of light to a smaller focal spot area in the diffraction limit of about $\lambda_n^2$. Second, the stable relative phase between harmonics enables the formation of attosecond intensity spikes with a duration much shorter than that of the initial optical cycle $T_0 = \lambda_L/c$. These two ingredients, namely, high orders and phase locking, are common to many harmonic sources. The third ingredient, however, is a unique feature of the SHHG mechanisms discussed here. The conversion efficiency must decay sufficiently slowly for many harmonic orders to be of comparable intensity in the CHF[5,10,15–18]. In the relativistic mirror approximation, the energy conversion $\eta_n$ has been shown to converge to $\eta_n = n^{-8/3}$ in the high-intensity limit, and, when combined with the $\lambda_n^2$ reduction in the source size, we immediately see that the harmonics decay as approximately $n^{-2/3}$ in intensity and $n^{-1/3}$ in field strength. Consequently, if the 'slow decay at high efficiency' condition can be met, many adjacent orders will have comparable field strength and contribute substantially to the ultimate field distribution in a CHF. The experimental results presented in Fig. 1

[1]Atomic and Laser Physics sub-Department, Clarendon Laboratory, Department of Physics, University of Oxford, Oxford, UK. [2]John Adams Institute for Accelerator Science, Department of Physics, University of Oxford, Oxford, UK. [3]Centre for Light Matter Interactions, School of Mathematics and Physics, Queen's University Belfast, Belfast, UK. [4]Central Laser Facility, Rutherford Appleton Laboratory, Didcot, UK. [5]AWE, Reading, UK. [6]Gérard Mourou Center for Ultrafast Optical Science, University of Michigan, Ann Arbor, MI, USA. [7]Helmholtz Institute Jena, Jena, Germany. [8]Institute for Optics and Quantum Electronics, Friedrich Schiller University, Jena, Germany. [9]GSI, Darmstadt, Germany. [10]These authors contributed equally: Robin J. L. Timmis, Colm R. J. Fitzpatrick, Jonathan P. Kennedy, Holly M. Huddleston, Elliott Denis, Mark Yeung, Brendan Dromey, Peter Norreys. ✉e-mail: robin.timmis@physics.ox.ac.uk; b.dromey@qub.ac.uk

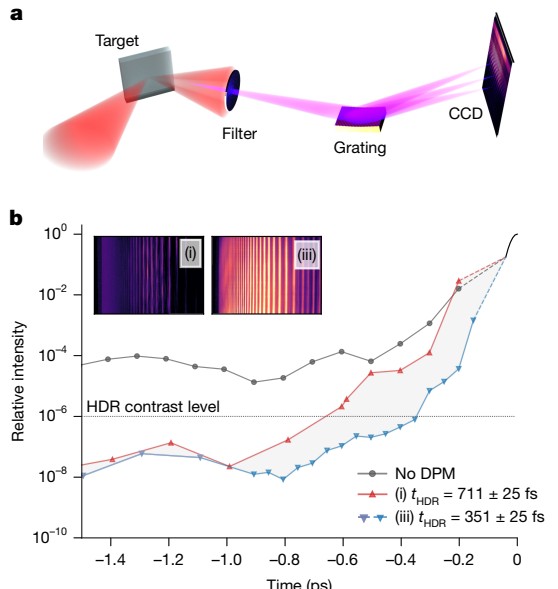

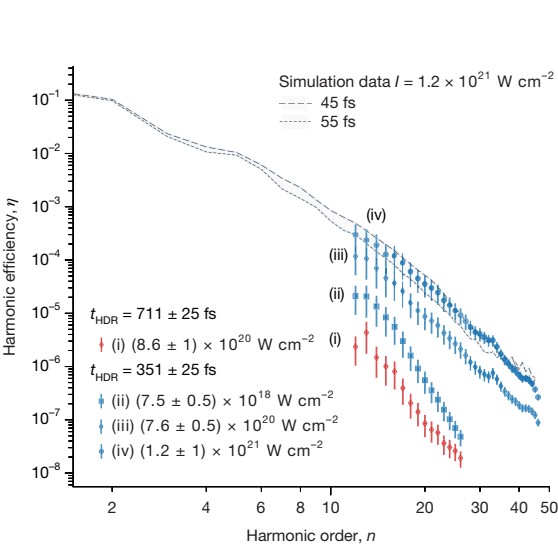

**Fig. 1 | Fine-tuning of laser pulse contrast for efficient XUV harmonic generation. a**, Schematic of the optical set-up for harmonic beam generation and diagnosis. The laser pulse is focused to intensities of $1.2 \times 10^{21}$ W cm$^{-2}$ by an off-axis parabola. The reflected radiation is filtered by an Al foil (up to 3 μm thick) and dispersed by a 300 line per mm flat-field grating onto a CCD. **b**, Third-order cross-correlation traces of the Gemini main beam for different DPM configurations in the −1.5 ps to 0 ps window (data resolving the $50 \pm 5$ fs pulse in the −0.1 ps to 0 ps window are taken from a separate frequency-resolved optical gating device). The high dynamic range rise time, $t_{HDR}$, is defined as the time taken to increase from $10^{-6}$ to the peak intensity. For a DPM set-up with $t_{HDR} = 711 \pm 25$ fs (**b**, red trace), the harmonic signal in the inset labelled (i) (raw CCD image) is observed. This corresponds to the red trace labelled (i) in **c**.

Enhancing the performance of the DPM to $t_{HDR} = 351 \pm 25$ fs (**b**, blue trace) produces a notably brighter signal on the CCD (inset labelled (iii)) for a similar on-target intensity as (i). This corresponds to the blue trace labelled (iii) in **c**. **c**, Experimental efficiencies for a range of parameters. For the optimal harmonic beam (labelled (iv) in **c**), the experimental data are compared with an equivalent 2D PIC simulation with $1.2 \times 10^{21}$ W cm$^{-2}$ and pulse durations of 45 fs and 55 fs under optimal conditions. In particular, this shot produced an XUV beam containing $9.5 \pm 4.9$ mJ in the interval between the 12th and 47th harmonic orders at an efficiency of $0.17 \pm 0.08\%$. Note that the inherent laser contrast (**b**, grey trace) results only in spectral line emission, which is indicative of an interaction with a considerably pre-expanded plasma with a long-density scale length.

confirm that this final important ingredient has now been achieved in the laboratory for SHHG.

As mentioned above, the first two ingredients required for CHF have been demonstrated previously, with temporal compression of the fundamental radiation observed in the specularly reflected direction[6,10,15,19–21] and 'denting' by the steep intensity gradients of the incident laser pulse[2,5] providing curved harmonic wavefronts in reflection that permit diffraction-limited performance in experiments[5,22–24]. Simulations predict that the combination of these properties and optimized generation conditions[2,3] can enable substantial intensity boosts in a CHF[1]. Despite early advances, however, progress towards a practical CHF slowed as the absolute conversion efficiencies predicted by theory for SHHG from petawatt (PW)-class laser-solid target interactions[25,26] could not be realized in the laboratory. This was due to several factors.

First, although the relative efficiency scaling slope of $n^{-8/3}$, consistent with theory, has been observed[16,17], these data were obtained for pulse durations >500 fs ($5 \times 10^{-13}$ s). (Note that in the closely related coherent synchrotron emission SHHG mechanism, the slope efficiency deviates[18]). For these time frames, the relativistic plasma evolves rapidly over the duration of the interaction (see the supplementary information in ref. 18). More recently, extensive numerical work[27,28] explored SHHG under a wide range of interaction conditions for single-cycle pulses to show how small changes to important parameters such as plasma density scale length or laser pulse intensity can alter the properties of the attosecond electron bunches at the target surface[29]. This, it is shown, can then lead to dramatic changes in the slope of efficiency scaling with $n$. The implication of this is clear; for pulses with many hundreds of optical cycles, the most likely outcome is that the optimum conditions are only fleetingly achieved during the interaction between the laser pulse and the evolving plasma. This 'windowing' on the interaction reduces the absolute conversion efficiency to only a

fraction of what theory predicts for (near-)constant conditions, despite harmonic-dependent efficiency scalings consistent with theory being observed in experiments.

Second, over the last decade, advances in PW-class lasers have made increasingly shorter pulse durations (<50 fs) available. This is beneficial as it should, in principle, optimize at the ultrasteep plasma density gradients outlined above. The drawback, however, is that there is less time for the laser pulse to modify and shape the evolving plasma density profile. Therefore, for experiments with limited control over the interaction conditions, in particular for PW-class systems, it is likely that the narrow window for optimized plasma-density profile formation is never reached, as this depends on the exact pulse shape of the laser over many orders of magnitude in intensity (the so-called pulse contrast).

In the laboratory, achieving the well-defined plasma surface outlined above typically requires using a double plasma mirror (DPM) system to tailor the native laser contrast ahead of the high-intensity interaction. Plasma mirrors are optical switches formed in the near-field of focusing laser beams, typically at intensities of $10^{14}$–$10^{15}$ W cm$^{-2}$ (refs. 30,31). These provide contrast enhancements extending up to $10^5$ by rapidly switching from low reflectivity off a 'cold' dielectric substrate to high reflectivity off a critical density plasma formed on this substrate by the rising edge of the incident laser pulse. Although DPM performance is well established on nano- and picosecond time frames[32–35], the factors that determine how rapidly the switching to high reflectivity occurs on sub-picosecond time frames are less clear and, in particular, the role that this 'near-time' DPM performance has on efficient SHHG.

## High-order harmonics from relativistic plasmas

An experiment to investigate the near-time contrast limits for efficient SHHG generation on PW-class laser systems was performed at

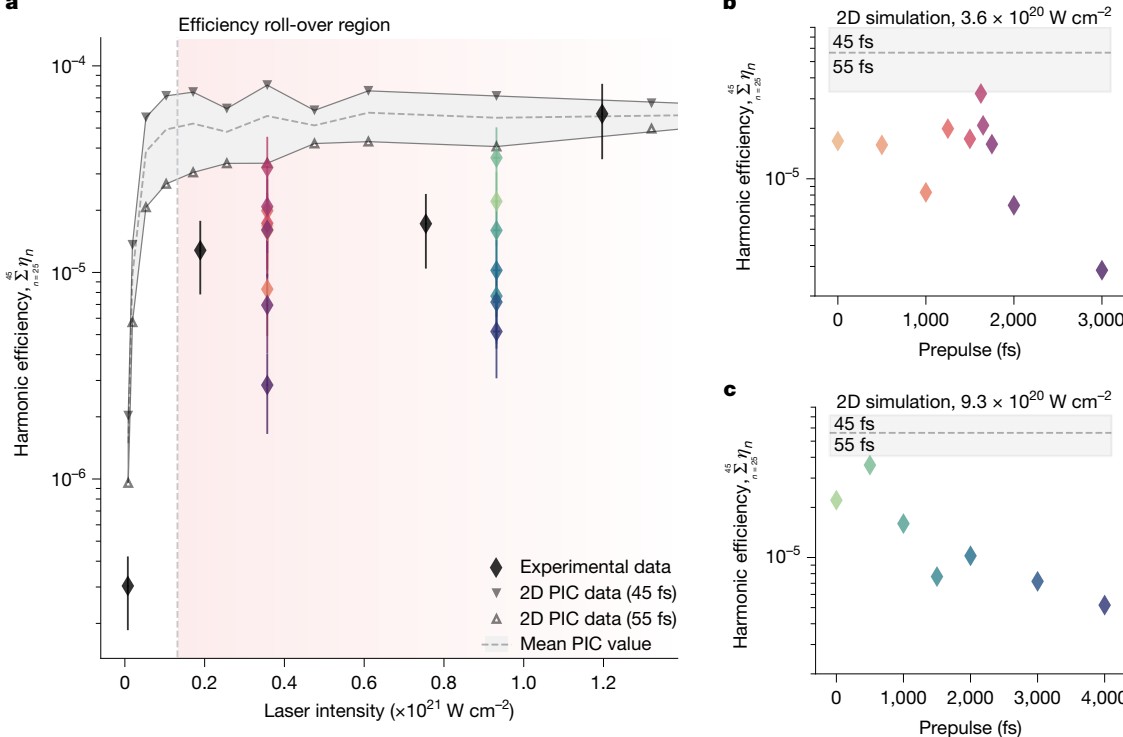

**Fig. 2 | Harmonic generation in the efficiency limit. a**, Comparison of experimentally measured XUV harmonic efficiencies obtained for a DPM configuration delivering $t_{HDR} = 351 \pm 25$ fs (black diamonds) with 2D PIC simulations (grey points and dashed line) as a function of incident laser pulse intensity. Simulations were optimized at each intensity by varying an exponentially decaying plasma density scale length between $0.12\lambda_L$ and $0.16\lambda_L$. At low intensity, simulations show that a rapid growth in harmonic efficiency integrated over harmonic orders 25–45 is expected before this increase levels off or 'rolls over'. This harmonic range is chosen to ensure coherent wake emission harmonics[44] are not contributing to the observed signal for the lower intensity interactions. The roll-over region is defined as the regime in which the relative increase in harmonic conversion efficiency with increasing intensity falls below 25%, the onset of which begins at $1 \times 10^{20}$ W cm$^{-2}$ (vertical dashed line).

In experiments, the maximum conversion efficiency is obtained only for an intensity of $1.2 \times 10^{21}$ W cm$^{-2}$. At lower intensities, the intrinsic prepulse is insufficient for efficient laser–plasma coupling, preventing optimum efficiency. Note that the bars correspond to the absolute uncertainty on the measurements. **b**, Introducing a prepulse using the method outlined in ref. 37 brings the efficiency back in line with simulation, allowing the harmonic efficiency to be optimized in a narrow 50 fs window for an on-target intensity $3.6 \times 10^{20}$ W cm$^{-2}$. **c**, For the higher intensity of $9.3 \times 10^{20}$ W cm$^{-2}$, the same optimization trend is observed, albeit for shorter delays. The colours of the points in **b** and **c** relate the prepulse timing scan to the same colour data points in **a**. Thus, the observation of near-constant harmonic efficiency over a broad intensity range indicates efficiency saturation.

the Central Laser Facility at the Rutherford Appleton Laboratory of UKRI-STFC using the Gemini laser system (see the Methods for details).

Harmonic emission was observed only with the DPM system, as the intrinsic contrast caused early plasma expansion. To characterize sub-ps switching, we define the high dynamic range rise time, $t_{HDR}$, as the interval for the pulse intensity to rise from $10^{-6}$ to its peak (Methods).

This level is marked in Fig. 1b by the horizontal, black dotted line. When using a DPM configuration that yields a $t_{HDR} = 711 \pm 25$ fs (Fig. 1b, red trace) with an on-target intensity of $I \approx 8.6 \times 10^{20}$ W cm$^{-2}$, a harmonic spectrum containing several µJ is observed (Fig. 1b, (i)). The measured efficiencies for each harmonic are shown in Fig. 1c as the red points (see the Methods for details on harmonic energy deconvolution). Harmonic emission above background was not observed for higher intensities using the $t_{HDR} = 711 \pm 25$ fs DPM configuration. This is an indication that the laser pulse contrast was not sufficiently high on ultrafast (sub-ps) time frames to permit the generation of plasma conditions suitable for SHHG.

Tuning DPM performance (see Methods and Extended Data Fig. 1) by controlling the breakdown time on the DPM surfaces[36] to deliver the blue trace in Fig. 1b, with $t_{HDR} = 351 \pm 25$ fs, however, yields a striking performance enhancement. For a reduction of about 360 fs in $t_{HDR}$, an increase in harmonic efficiency of several orders of magnitude is observed (Fig. 1b, (iii)) for a similar on-target intensity as in Fig. 1b (i) ($I \approx 7.6 \times 10^{20}$ W cm$^{-2}$). Efficiencies for this contrast setting and a range

of intensities are shown by the blue points in Fig. 1c. At the highest intensity of $I \approx 1.2 \times 10^{21}$ W cm$^{-2}$, Fig. 1c (trace (iv)), for the interval from the 12th to the 47th harmonics inclusive, the reflected harmonic beam contains $9.5 \pm 4.9$ mJ at an efficiency of $0.17 \pm 0.08$%. The impact of the reduced $t_{HDR}$ is readily apparent when we note that for the blue trace contrast and an on-target intensity of only $I \approx 7.5 \times 10^{18}$ W cm$^{-2}$, a harmonic beam with approximately an order of magnitude higher conversion efficiency (Fig. 1c, trace (ii)) is observed when compared with that for the red trace contrast with a much higher intensity of $8.6 \times 10^{20}$ W cm$^{-2}$ (Fig. 1c, trace (i)).

Figure 1c also compares the experimental data with the absolute conversion efficiencies extracted from two-dimensional (2D) particle-in-cell (PIC) simulations optimized for the known experimental conditions (simulation parameters are listed in the Methods). In Fig. 1c, excellent agreement across three orders of magnitude of individual harmonic efficiency is found between experimental data ($I \approx 1.2 \times 10^{21}$ W cm$^{-2}$) and simulations performed at the same intensity. Overall, this shows that tuning the DPM performance to initiate a faster rise time ($t_{HDR} = 351 \pm 25$ fs) for the interaction enables the efficient coupling of laser pulse energy into the coherent harmonic beam using PW-class systems, in agreement with theory and simulation.

Although there is close agreement between optimized 2D simulations and experiments for focused intensities of $I \approx 1.2 \times 10^{21}$ W cm$^{-2}$,

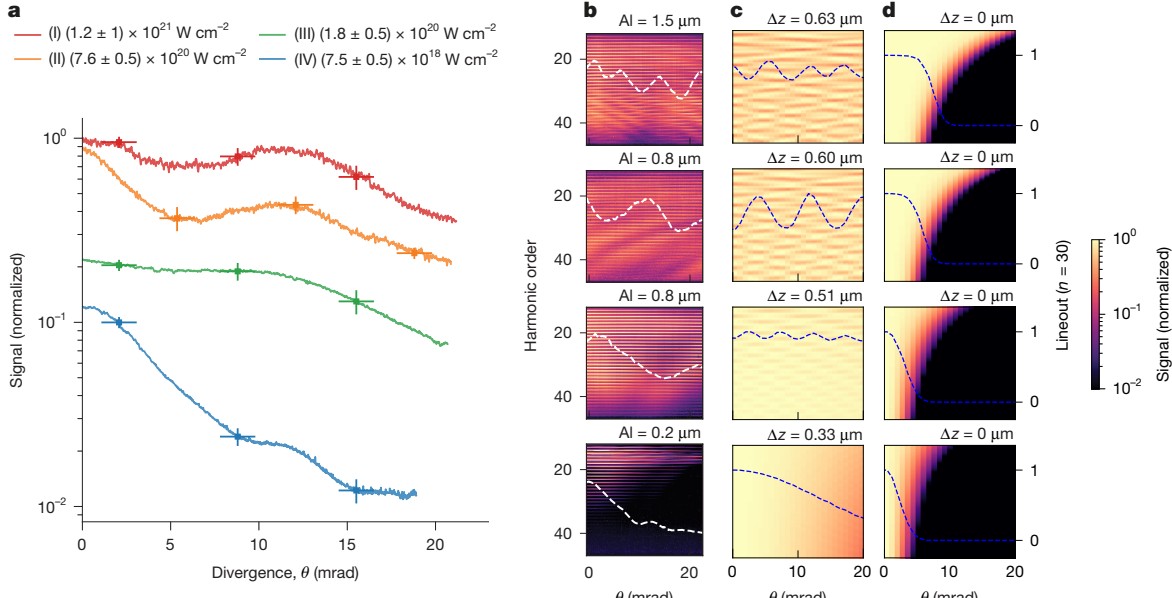

**Fig. 3 | Measured XUV harmonic beam profile dependence on laser pulse parameters and the development of spectrospatial modulations. a**, Experimentally measured harmonic beam spatial profiles summed over the harmonic range 30–35 inclusive. **b**, The corresponding CCD images, with the thickness of the Al filter for optical radiation used for each intensity given above the respective panel. The data are not corrected for filter transmission. The interaction intensity (given in I–IV) was varied by changing the width of the driving beam using an apodizer. As the apodizer size increases, the energy in the laser beam increases, and the laser focal spot size reduces, leading to a rapid intensity scaling for these experiments. An apodizer is used to maintain DPM performance (this requires non-varying near-field energy fluence for all experimental conditions). The colour scale in **b** has been chosen to highlight the features of interest. **c**, Applying the theories of relativistic spikes and ponderomotive deformation of plasma surfaces (denting) to the far-field diffraction pattern of the laser pulse spot for an incident top-hat profile beam and propagation back to the near-field for the harmonic emission qualitatively reconstructs the observed harmonic beam profiles observed in **a** and **b**. Dent depths, $\Delta z$, are calculated at the temporal peak of the laser pulse intensity assuming a Gaussian temporal profile. See the Methods for calculation details. For high intensities, plasma denting by the driving laser pulse leads to the emergence of interference fringes in the angular distribution. **d**, In the absence of plasma denting, increasing the laser pulse intensity yields a closer reconstruction of the top-hat near-field profile of the incident laser pulse with no strong interference pattern observed. The blue and white dashed traces track the beam profiles for the 30th harmonic order in each image for **b**–**d**.

Fig. 2a shows how this interaction is de-optimized for lower intensities, that is, the data (black diamonds) no longer match the maximum conversion efficiencies predicted by simulation (grey points and dashed line). This behaviour is attributed to a shorter plasma density scale length being generated by the proportionally reduced intrinsic prepulse intensity of the main laser pulse, that is, the native prepulse and temporal pedestal of the laser system before contrast enhancements. To confirm this, it was demonstrated how optimized conditions are returned by introducing an independent 50 fs prepulse with approximately $10^{-3}$ of the focused intensity of the main beam[37]. By fine-tuning the delay on this prepulse, it is possible to recover the numerically predicted conversion efficiencies for lower intensity interactions. For $I \approx 3.6 \times 10^{20}$ W cm$^{-2}$, Fig. 2b shows a narrow window (about 50 fs) within which the optimum sits, demonstrating a tightly defined range for optimum laser–plasma coupling during these high-intensity conditions. Similarly, Fig. 2c shows that even for a considerably higher on-target intensity of $I \approx 9.3 \times 10^{20}$ W cm$^{-2}$, a 200 ± 100 fs window for optimization persists close to the interaction of the main pulse. Importantly, it was not possible to make marked improvements (using this prepulse configuration) to the harmonic efficiencies observed at either the highest or the lowest intensity interactions shown in Fig. 2. For the highest intensity, $I \approx 1.2 \times 10^{21}$ W cm$^{-2}$, this is straightforward to understand as the signal is already optimized, in agreement with predictions from numerical simulations for the experimental conditions. By contrast, for the lowest intensities, $I < 1 \times 10^{19}$ W cm$^{-2}$, the reason why the predicted efficiency is not realized is more subtle, and is directly linked to the rapid efficiency scaling for intensity in this range. This is better understood by examining the evolving divergence of the harmonic emission as the interaction intensity is varied.

## Harmonic beam structure

Figure 3a traces a rapid increase in overall harmonic beam divergence (full-width at half-maximum (FWHM)) for increasing intensity, with divergence for harmonics in the range 30th–35th increasing from 5 ± 1 mrad for $I \approx 7.5 \times 10^{18}$ W cm$^{-2}$ to >35 ± 5 mrad for $I \approx 1.2 \times 10^{21}$ W cm$^{-2}$. Although some growth in divergence is anticipated because of the smaller spot sizes and larger divergence of the driving laser pulse in the higher intensity cases (see Fig. 3 caption and Methods), the striking deviation from a Gaussian-like profile has not been previously reported. Moreover, Fig. 3b also shows the presence of strong spectrally dependent modulations in the angular direction of the reflected beam profile (termed 'spectrospatial modulations'). Both these observations indicate a rising complexity in the interaction, which is directly related to the generation mechanism.

This complexity is investigated using a simple model that applies efficiency scaling from the theory of relativistic spikes[10] across the spatially varying intensity profile of the laser focal spot (full details on this model are provided in the Methods). In high-power laser systems, energy extraction from the amplification chain is maximized using a super-Gaussian, or top-hat, laser beam profile. This results in a focal spot distribution (Extended Data Fig. 2) with a central maximum intensity (peak) and radially symmetric 'wings' at the 5–10% intensity level of the peak (more rigorously, this can be approximated by a jinc function). These wings correspond to the higher spatial frequency components of the expanded laser beam. At the same time, the theory of relativistic spikes predicts a cut-off harmonic order that scales strongly with driving intensity[10]. Above this cut-off harmonic, there is a rapid drop in efficiency. For peak focused intensities $I < 1 \times 10^{19}$ W cm$^{-2}$

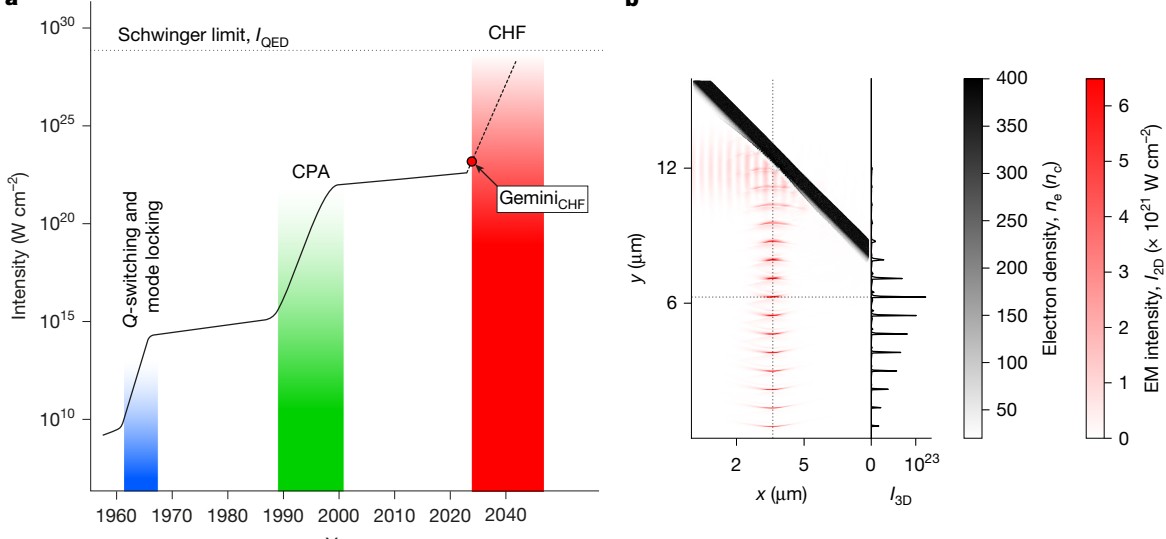

**Fig. 4 | Reaching extreme fields with CHF. a**, The evolution of achievable laser intensities is shown as a function of time since the development of the laser. It is characterized by large jumps achieved by new techniques allowing laser energy to be delivered into smaller volumes, with the shallow slopes in between corresponding to increasing the laser energy using a given technique. Most recently, CPA has resulted in a large increase. The spatiotemporal compression possible due to CHF is predicted to lead to a further increase. **b**, A snapshot of a 2D PIC simulation of the Gemini laser pulse interacting with an initially planar solid density plasma. The incoming laser pulse propagates in the $+\hat{x}$-direction.

The spatially varying laser pressure dents the plasma surface resulting in curved wavefronts in the specularly reflected harmonic beam, focusing the emission to high intensities. The peak focus (CHF) is marked by the vertical dashed line. By applying the methodology detailed in the Methods, the electromagnetic field intensity in the 2D simulation, $I_{2D}$, can be scaled to predict the intensity from such a surface curvature in 3D, $I_{3D}$. The CHF for optimized Gemini conditions, Gemini$_{CHF}$, is predicted to produce a gain of more than 80 times the incident intensity, corresponding to a maximum focused intensity of more than $1 \times 10^{23}$ W cm$^{-2}$ (red dot in **a**).

(here classed as mildly relativistic, see Methods for a discussion on driving laser intensity and its link to relativistic velocities for electrons), these two properties combine in the laboratory to produce a strongly intensity-dependent spatial filtering effect for harmonics in the interval from 25 to 45. The steep efficiency scaling for lower peak intensities (Fig. 2a) implies that the intensity in the wings is too low for efficient generation in this harmonic interval. This effectively prevents any high spatial frequency components from the driving laser pulse being imparted onto the emitted harmonic beam through the coherent up-conversion process, resulting in emission only from the central maximum leading to a Gaussian-like harmonic emission cone. Figure 3d shows the intensity-dependent evolution of the harmonic beam divergence for a laser pulse with a top-hat beam profile assuming generation from a flat plasma surface. Although the anticipated Gaussian-like profile due to spatial filtering of the central maximum is demonstrated for $I < 1 \times 10^{19}$ W cm$^{-2}$, as $I$ increases into the efficiency roll-over regime (that is, the intensity range to the right of the dashed vertical line in Fig. 2a), the higher spatial frequency components from the wings of the focal spot begin to contribute to the generation process. Tracking the lineouts for the 30th harmonic in Fig. 3d, as intensity increases, the incident top-hat (super-Gaussian) intensity distribution is increasingly reproduced in the harmonic beam, undoing the spatial filtering effect (Gaussian-like profile) observed at lower intensities.

Figure 3c demonstrates that an increase in plasma surface dent depth results in a rapid growth in harmonic divergence with intensity (much greater than that expected for a flat generation surface), along with the emergence of strong spectrospatial modulations in the beam profile. See the Methods for a more detailed discussion on the plasma surface evolution during the interaction. These modulations can be achieved only in experiment if the conversion efficiency is comparable across the entire focal spot—that is, in the efficiency roll-over regime variations in harmonic signal scale with energy. The observation in experimental data of an intensity-dependent departure from Gaussian-like beams

and the emergence of strong modulations, therefore, provides further evidence that the results have entered the efficiency saturation regime for the highest intensity interactions. Simulations (Fig. 2a, grey dashed line) show that efficiency roll-over sets in for $I > 1 \times 10^{20}$ W cm$^{-2}$, above which the rapid growth in efficiency with intensity flattens off. There is a transition in the experimentally observed beam structure at this point with pronounced spectrospatial modulations only for intensities well above this roll-over threshold (Fig. 3b, I and II). This is an important observation as it demonstrates that conditions for efficient harmonic generation in the presence of curved plasma surfaces are present here across the full width of the focal spots, a key requirement for obtaining a CHF in the laboratory.

## Conclusions

Figure 4 shows the predicted CHF field strengths within the historical progression from Q-switching and mode-locking to Chirped Pulse Amplification (CPA), highlighting its potential as the next major advance. Under efficiency-optimized conditions, the predicted intensity gain in the CHF scales rapidly with driving laser pulse intensity— $I_{CHF}/I \propto a_0^3$, where $a_0$ is the normalized vector potential for the driving laser pulse (see the Methods for details on $a_0$). The potential has already been discussed in the literature with applications to photon–photon scattering[38] and possible experimental configurations[39–41]. Reaching these high intensities without CHF requires considerably larger laser systems, such as Extreme Light Infrastructure–Nuclear Physics (ELI-NP)[42]. ELI-NP has identified SHHG as a promising tool for generating bright XUV beams for Strong-Field Quantum Electrodynamic (SF-QED) studies[43]. Applying the $I_{CHF}/I \propto a_0^3$ scaling to 10 PW systems such as ELI-NP, 20 PW lasers such as the Vulcan 20-20 and NSF-OPAL laser at University of Rochester, as well as the 50 PW SEL facility in Shanghai, these lasers can, in principle, already access intensities approaching $10^{29}$ W cm$^{-2}$, provided the efficiency roll-over regime described in this work is realized and the approximations made in the

simulations remain valid. The preconditions are now met to investigate the extreme intensity regime that is predicted to be possible using CHF. Current and next-generation multi-PW facilities enable these studies but require improved plasma control, including high-contrast, fast rise times and tunable density gradients and curvature.

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

## Methods

### Experimental details

The experimental results presented in this paper were obtained using the Gemini laser system. A DPM system is used to improve the laser contrast to $I_{max}/I(t) > 10^8$ at times more than 1 ps before the peak of the pulse, whereas the sub-ps contrast is discussed in the text referring to Fig. 1. A total throughput of 50% was measured, which leads to on-target pulse energies of 5 J in the $50 \pm 5$ fs duration pulses with $\lambda_L = 800$ nm, which are focused by an $f/2$ parabola onto a polished fused silica target.

Pulses with energies of up to 12 J (before DPM system) in $50 \pm 5$ fs at a central wavelength of 800 nm were used, which, when focused to a FWHM spot size of 2 μm, reach peak intensities $I > 10^{21}$ W cm$^{-2}$. As shown in Fig. 1a, these were focused onto optical-grade fused silica targets in p-polarization at an incidence angle of 45°, and the spectrum of extreme ultraviolet radiation emitted in the direction of specular reflection was recorded.

The on-target intensity was varied by apodizing the beam, which both reduces the laser pulse energy and increases the focal spot size but maintains the same near-field intensity so that the DPM response and contrast are unchanged. The reflected harmonic beam was detected using a cylindrically curved XUV flat-field spectrometer consisting of a 300 lines per mm grating imaging the source in the spectral dimension. Aluminium filters with thicknesses ranging from 0.2 μm to 3 μm were used to attenuate optical emission. No focusing optic was used so that the XUV signal is incident directly onto the charge-coupled device (CCD). The harmonic spectra were detected using a back-thinned ANDOR CCD (Andor DV436) with a resolution-limited pixel size of 13.5 μm placed 1.2 m from the interaction point.

The plasma density gradient was controlled by a 25 mm diameter, 3 mm thick fused silica substrate with an anti-reflection coating on the front side and high reflectivity on the rear. This pick-off mirror was inserted into the main beam line in front of the last mirror before the parabola. This introduced a prepulse, which is focused by the same parabola as the main beam onto the target but with a lower intensity because of the larger focal spot size. Precise adjustment of the distance between the substrate and the full-beam mirror allowed for prepulse timing to be controlled to within 25 fs of the main pulse. This fine timing control enabled accurate tailoring of plasma expansion before the arrival of the driving pulse.

Laser contrast enhancement was achieved through measurements that determined that the anti-reflection coating on the first plasma mirror (PM) of the DPM was breaking down too early in the rise time of the native (unaltered) Gemini pulse contrast. This resulted in the 'slow rise time' DPM configuration ($t_{HDR} = 711 \pm 25$ fs; Fig. 1b, red trace). By replacing the first PM with an uncoated substrate, we improved the DPM performance to have a $t_{HDR} = 351 \pm 25$ fs (Fig. 1b, blue trace). The motivation for making this change comes from a separate branch of study on ultrafast materials science that focuses on the part played by materials that are highly structured on the nanoscale in the lifetime of excited electrons before material breakdown[45,46]. Note that for different peak intensities, $t_{HDR}$ will describe different absolute intensities, whereas the ratio remains the same. This should, therefore, be considered carefully in the context of a given peak intensity interaction.

### Harmonic energy deconvolution

To obtain the overall efficiency, the spectral response as a function of wavelength, $\lambda$, of all components of the spectrometer were accounted for separately as indicated by

$$S(\lambda) = \text{Al} \times \text{QE} \times G \times \text{Al}_2\text{O}_3 \times \text{CH}, \tag{1}$$

where Al is the aluminium filter transmission[47] (0.2–3 μm), QE is the quantum efficiency of the back-thinned Andor DV436 (ref. 48), G is the calculated efficiency of the SHIMADZU-L0300-20-80 flat-field grating[49], $\text{Al}_2\text{O}_3$ is the contaminant aluminium oxide layer present on the filters (see aluminium oxide layer calibration below) and CH is the hydrocarbon contaminant layers[47].

The energy per count is calculated as

$$E(\lambda) = \frac{G \times S(\lambda) \times \varepsilon \times q_e}{\rho_{frac}} \tag{2}$$

where $G = 2$ is the number of electrons per count (e$^-$ per count), $\varepsilon$ is the average energy required to produce an electron–hole pair in silicon, 3.65 eV, $q_e$ is the electronic charge and $\rho_{frac}$ is the fraction of the beam incident onto the CCD. This is restricted by the angle of acceptance of the flat field grating of 3.66 mm and the camera chip width of 27.6 mm. We assume a two-dimensional (2D) Gaussian beam propagating along the $z$-axis, with its transverse intensity profile described by

$$G(x,y) = \exp\left(-\left[\frac{(x-x_0)^2}{2\sigma_x^2} + \frac{(y-y_0)^2}{2\sigma_y^2}\right]\right), \tag{3}$$

where $(x_0, y_0)$ is the centre of the beam and $\sigma_x$ and $\sigma_y$ are the standard deviations of the beam along the $x$- and $y$-directions, respectively (measured values of beam divergence are used). We need to compute the fraction of the total beam that falls within a rectangular aperture defined by

$$x \in [x_{min}, x_{max}], y \in [y_{min}, y_{max}].$$

This fraction is given by the ratio

$$\rho_{frac} = \frac{\int_{y_{min}}^{y_{max}} \int_{x_{min}}^{x_{max}} G(x,y)\,dx\,dy}{\int_{-\infty}^{\infty} \int_{-\infty}^{\infty} G(x,y)\,dx\,dy} \tag{4}$$

The total Gaussian beam over all space is $2\pi\sigma_x\sigma_y$ thus,

$$\rho_{frac} = \frac{1}{4}\left[\text{erf}\left(\frac{x_{max} - x_0}{\sqrt{2}\,\sigma_x}\right) - \text{erf}\left(\frac{x_{min} - x_0}{\sqrt{2}\,\sigma_x}\right)\right] \\ \times \left[\text{erf}\left(\frac{y_{max} - y_0}{\sqrt{2}\,\sigma_y}\right) - \text{erf}\left(\frac{y_{min} - y_0}{\sqrt{2}\,\sigma_y}\right)\right] \tag{5}$$

By measuring the beam divergence (Fig. 3), we then calculate the associated fraction $\rho_{frac}$ of the beam that was incident onto the CCD. The uncertainty in the energy deconvolution is dominated by systematic uncertainty in the thickness of surface oxide ($\text{Al}_2\text{O}_3$), carbon contamination layers (CH) and the beam divergence measurement ($\rho_{frac}$). The correction is estimated by evaluating it at two extreme limits and then taking half the difference between the resulting corrected values as a symmetric uncertainty bound.

### Aluminium oxide layer calibration

A 2.15 μm thick aluminium foil, which was used as a filter for the harmonic signal, was imaged at the Ewald Microscopy Centre, Queen's University Belfast, to determine the thickness of the oxide containment layers. A cross-sectional lamellae was prepared using a Tescan focused ion beam scanning electron microscope (SEM) Lyra3, which can be seen in Extended Data Fig. 3a. High-angle annular dark-field imaging was performed, which images the front surface of the aluminium foil. Moreover, energy-dispersive X-ray (EDX) mappings were performed on this section of the lamellae, exhibiting a difference in the layer structure. Elemental mapping was used across the foil to identify the oxygen region corresponding to the aluminium oxide formed on the surface. The thickness of the oxide layers was observed to vary between 7 nm and 16 nm as seen in Extended Data Fig. 4b. This is consistent

with the native oxide typically formed on aluminium exposed to ambient conditions. However, it differs from other studies that measured the oxide layer thickness as 8 nm in total, not just the front surface thickness as measured here[50]. This variation of oxide layers has been included in the experimental uncertainties in the presented energy calculations. The need for gold sputtering of the sample surface and the low signal yield from light elements make it difficult to measure the hydrocarbon contaminant layer on the sample; we refer to ref. 50 for this value.

## Grating second and third orders

Using a diffraction grating to image XUV harmonics, the appearance of features at what seem to be 'half-order' or 'third-order' harmonic positions is attributed to the higher diffraction orders of the grating appearing with the first-order signal. In Extended Data Fig. 5a, the lower-order harmonic region of the XUV spectrum between 47 nm and 45 nm, there are apparent 'half-order' harmonics observed. However, when comparing these features to the higher-order harmonics shown in Extended Data Fig. 5b between 20 nm and 22.5 nm, it can be seen in the magnified region of the plots that the half harmonic order that is seen to be at 19.5th order is actually the 39th-order harmonic being imaged in second order because of the matching spatial and spectral shape of these two images.

This highlights the importance of accounting for second- and even third-order diffraction contributions when analysing and calculating the total energy distribution in the lower-order spectral region. Using an aluminium filter allows transmission of certain XUV wavelength ranges while blocking others. By choosing a filter that transmits only the desired harmonic range and blocks shorter wavelengths, we can suppress unwanted second- or third-order contributions. However, using a 300 l mm$^{-1}$ grating in the wavelength range of 80–20 nm, signals close to the aluminium L-edge cut-off can appear in the second and third order with the lower-order harmonics.

In Extended Data Fig. 6a, the ratio of the first and second diffraction orders of the measured SHIMADZU-LO300 flat-field grating is shown[49]. This was measured using the laser-driven high harmonics that were incident onto a pinhole. It is seen in Extended Data Fig. 6a that the second-order efficiency approaches the first-order efficiency at higher frequencies. In Extended Data Fig. 6b, the ratio of the first and third diffraction orders are shown, which has a similar trend in approaching the first order when moving towards higher frequencies.

A logistic function was fitted to the data

$$R_{\frac{S_{2,3}}{S_1}}(n) = \frac{v}{1 + e^{-k(n-n_0)}} + b, \tag{6}$$

where, for the second diffraction order, $v = 0.92$, $n_0 = 40$, $k = 0.37$ and $b = 0.12$ and for the third diffraction order, $v = 0.79$, $n_0 = 44$, $k = 0.44$ and $b = 0.13$. Failing to account for these effects leads to overestimating the reflected harmonic energy in the lower orders, for which the aluminium filter attenuates the spectrum considerably. The trend given by equation (6) was used to account for the second- and third-order diffraction contributions. Owing to this overlap being unavoidable, using the grating response, we can deconvolve the overlapping orders from the spectrum.

## Numerical simulations

The SHHG interaction was modelled using high-resolution 2D simulations performed with the PIC code Smilei[51]. A spatial resolution of 512 cells per laser pulse wavelength and a temporal resolution of 1,024 timesteps per laser pulse cycle were used, enabling the resolution of harmonics beyond the aluminium L-edge (17 nm) (ref. 28). The p-polarized laser pulse is incident on a fully ionized SiO$_2$ target of number density $6.62 \times 10^{23}$ cm$^{-3}$ with an exponential preplasma ramp. Optimal preplasma scale lengths for XUV SHHG efficiencies were

determined from one-dimensional (1D) PIC simulations and ranged from $0.12\lambda_L$ to $0.16\lambda_L$. XUV SHHG efficiencies are typically optimized for scale lengths in the range of $0.1$–$0.2\lambda_L$ (refs. 23,37,52). Particles are initialized with 100 macro-electrons per cell at a temperature of 115 eV and 4 macro-ions per species per cell at zero temperature. The laser pulse is modelled as a spatiotemporal Gaussian with a spatial FWHM of 2 µm and a temporal FWHM of 45 fs and 55 fs. As the pulse duration is anticipated to influence the attainable absolute efficiencies in the specularly reflected harmonic cone, simulations were performed to bound the range sampled in the experiment due to shot-to-shot variations (±5 fs), therefore yielding a range of optimized efficiency values (Fig. 1c, grey shaded area). Silver Müller boundary conditions allow the free movement of particles and electromagnetic fields into and out of the simulation window[53]. Smilei's in-built Bouchard solver[54] is applied to reduce the error from numerical dispersion in 2D inherent to traditional finite difference solvers[55]. It has been shown that modifications to the finite difference approach can sufficiently reduce this error[56].

## Normalized vector potential

It is standard practice in PIC simulations to define the normalized vector potential, $a_0$, instead of intensity. For a laser pulse of frequency, $\omega$, and peak electric field amplitude, $E$, corresponding to an intensity, $I = \frac{1}{2}c\epsilon_0 E^2$, the normalized vector potential is expressed as $a_0 = \frac{eE}{m_e c\omega}$, where $c$ is the speed of light, $\epsilon_0$ the vacuum permittivity and $m_e$ the mass of an electron. In particular, the onset of relativistic effects, including SHHG, occurs for $a_0 \geq 1$. In Fig. 1c, an $a_0$ of 24 is used in the PIC simulation to model the interaction. In Fig. 2, $a_0$ values ranging from 2 to 25 are used to explore the parameter space and in Fig. 2b,c, the $a_0$ values are 13 and 21, respectively. An $a_0$ of 25 is used in Fig. 4b.

## Analytical model of XUV beam profiles

In reality, the plasma surface is not expected to remain flat over the entire duration of the Gemini laser pulse. The steep intensity gradients result in a ponderomotive force, initially deforming the plasma surface into a concave 'dent' that follows the profile of the central maximum of the focal spot. The dent depth $\Delta z$, defined as the difference between the position of the critical density surface at the spatiotemporal peak of the driving laser pulse and its original position, is calculated from an analytical model[22] detailed below for the experimental conditions and compared with 2D PIC simulations with the same parameters. An intensity-dependent phase term is included in the far-field harmonic beam profile model to account for this denting.

The Fraunhofer diffraction equation describes the propagation of a monochromatic, $\lambda$, scalar field amplitude to large distances as a Fourier transform. It describes the propagation of a focused laser pulse from its far-field profile, $U_0(x', y')$, to its near-field profile, $U(x, y, z)$, as

$$U(x, y, z) \sim \iint U_0(x', y') e^{-2\pi i(f_x x' + f_y y')} dx' dy'$$
$$\sim \mathcal{F}(U_0(x', y'))|_{f_x = x/\lambda z, f_y = y/\lambda z}, \tag{7}$$

where $z$ is the distance from the focal spot. The far-field $n$th harmonic spatial profile is modelled by applying the theory of relativistic spikes[10]. Writing this adjustment as $S = S(n, a_0(x', y'))$, which contains a sharp roll-off at a harmonic order that scales as $a_0^3$, the far-field amplitude of the $n$th harmonic is

$$U_0(x', y', n) \sim \sqrt{S(n, a_0(x', y'))}\, U_0(x', y'). \tag{8}$$

Propagating the far-field harmonic beam profile to the near-field,

$$U(x, y, z, n) \sim \mathcal{F}(U_0(x', y', n))|_{f_x = x/\lambda_n z, f_y = y/\lambda_n z} \tag{9}$$

where $\lambda_n$ is the wavelength of the $n$th harmonic.

Radiation pressure of the laser pulse on the target surface causes a curvature of that surface, $\Delta z = \Delta z_i + \Delta z_e$, composed of the motion of the ion profile, $\Delta z_i$, and the excursion of the electrons, $\Delta z_e$. This curvature is modelled with the addition of a phase term, $\phi_n = 2k_n\Delta z\cos\theta$, to the far-field profile. A previous study provided a model for $\Delta z$ (ref. 22). For the Gemini pulse duration, the surface curvature varies slowly in time around the peak of the laser pulse, at $t_p$, when most harmonic emission occurs. The ion surface dent produced by a laser pulse with normalized vector potential $a_L(x', y', t')$ incident at an angle $\theta$ on an exponential preplasma of scale length $L$ is

$$\Delta z_i(x', y') = 2L\ln\left(1 + \frac{\Pi_0}{2L\cos\theta}\int_{-\infty}^{t_p} a_L(x', y', t')\mathrm{d}t'\right) \quad (10)$$

where $\Pi_0 = (RZm_e\cos\theta/2AM_p)^{1/2}$; $R$ is the reflectivity of the relativistic plasma mirror; $Z$ and $A$ are, respectively, the average charge state and mass number of the ions; $m_e$ is the electron mass and $M_p$ is the proton mass. The electron excursion is

$$\Delta z_e(x', y') = 2L\ln\left(1 + \frac{2a_L(x', y')(1+\sin\theta)}{2\pi(\cos\theta)^2 L/\lambda}e^{-\Delta z_i/L}\right). \quad (11)$$

Extended Data Fig. 7 compares the analytically calculated dent from ion motion to 2D PIC simulations with laser pulse intensities consistent with the shots of Fig. 3. Reflectivities are extracted from the simulations. Accounting for the corresponding phase term, the near-field intensity profile of the $n$th harmonic is

$$I(x, y, z, n)$$
$$\sim \left|\mathcal{F}\left(\sqrt{S(n, a_0(x', y'))}\, U_0(x', y')e^{-2ik_n\Delta z(x', y')\cos\theta}\right)\big|_{f_x = x/\lambda_n z, f_y = y/\lambda_n z}\right|^2. \quad (12)$$

We note that additional contributions to harmonic focusing may arise from intensity-dependent shifts of the apparent reflection point predicted by relativistic similarity theory[57,58]. Overall, this modelling demonstrates that, as well as increased energy in the harmonic beam, the experimental demonstration of the theoretically efficiency-optimized regime is also confirmed by (1) an intensity-dependent reduction in spatial filtering (departure from Gaussian-like beam profile); (2) a rapid, intensity-dependent growth in divergence; and (3) strong spectrospatial modulations in the emitted harmonic beam. The second and third points are also indications of efficient emission from a concave-dented plasma.

Although a CHF was not measured directly in this work, the observation of rapid increases in beam divergence with intensity, accompanied by the emergence of strong spectrospatial modulations provides evidence for the generation of CHF-suitable conditions in this experiment. This is further supported by 2D simulations, which predict the formation of a CHF during the experiment (Fig. 4). In the future, some degree of independent control of the plasma surface curvature is necessary to optimize both SHHG and the CHF performance simultaneously. This control could be passive, that is, a pre-shaped target that compensates for induced curvature, or active—that is, use of a substantially longer prepulse in conjunction with a single or few-cycle high-power driving laser pulse to exert absolute control over the shape of the surface at the moment of generation.

It remains uncertain how the spectrospatial modulations observed in this work will interplay with the spatiotemporal couplings that become increasingly more impactful at multi-PW levels, but techniques are now available for on-shot quantification of these couplings[59]. Moreover, although spatiotemporal laser–plasma couplings can play a substantial part in the quality of a CHF, in the efficiency limit, simulations show that these are dominated, and so can be controlled, by the driving field of the incident laser pulse[3,60].

## CHF 3D gain convergence

The CHF gain in 3D was estimated using 2D PIC simulations, following the methodology in ref. 39. The electromagnetic field intensities of the reflected harmonic beam are extracted from 2D PIC simulations at the time of generation of the brightest CHF, $I_f$, and at the time of generation of the attosecond pulse that produces this CHF, $I_0$. The snapshot of Fig. 4b is taken at the time of generation of the brightest CHF. The 3D boost can now be calculated. First, the gain from temporal compression, $\Gamma_D$, is $I_0/I_L$, where $I_L$ is the intensity of the incident laser pulse. The 2D gain from spatial compression is $\Gamma_{2D} = I_f/I_0$. The 3D gain that can be anticipated from spatial compression is $\Gamma_{3D} = \Gamma_{2D}^2$ and the full 3D gain is $\Gamma = \Gamma_D\Gamma_{3D}$. The attosecond pulses of Fig. 4a are then scaled using these equations and making the approximation that temporal gain is constant. As high harmonic orders contribute substantially to the intensity at the CHF, the accurate measurement of gain requires an unfeasible simulation resolution. To test the quality of the resolution used for the analysis, a scan of gain as a function of simulation resolution is plotted in Extended Data Fig. 8. A lower bound on the gain of 88 is extracted from the highest accessible resolution simulation, but higher gains seem probable.

## Data availability

The datasets generated during and/or analysed during this study are available from the corresponding authors upon reasonable request.

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

**Acknowledgements** We acknowledge the contributions of the CLF staff, in particular A. Thomas, conversations with M. Zepf's group and the Smilei developers for their assistance with simulations. This work used the ARCHER2 UK National Supercomputing Service (https://www.archer2.ac.uk) through the EPSRC HEC grant (EP/X035336/1). The thin-film analysis was performed by the Ewald Microscopy Facilities in the School of Mathematics and Physics at Queen's University Belfast. This work was funded by the EPSRC HEC grant (grant nos. EP/X035336/1, EP/W017245/1, EP/P010059/1 and EP/P016960/1), the AWAKE2 grant (ST/X005518/1), the JAI grant (ST/V001655/1), the Oxford-Living Optics and Oxford-IBM Computational Discovery grants, the Oxford Clarendon Scholarship scheme, the Deutsche Forschungsgemeinschaft (DFG, German Research Foundation; grant no. 392856280) and the National Science Foundation under award 2126181.

**Author contributions** R.J.L.T. led the experimental campaign and data analysis, assisted by C.R.J.F., J.P.K., H.M.H. and E.D.; P.N. was the overall principal investigator supervising the project with strong collaborative support from B.D. and M.Y.; M.Z., E.G., K.K. and P.P.R. provided useful insights. A.J., C.B., D.S., D.M., E.A., H.M., J. Rebenstock, J.N., J.L., J. Redfern, N.B., O.F., R.R., S.A., S.H. and Z.Z. contributed to the experiment, data analysis and writing of the paper.

**Competing interests** The authors declare no competing interests.

**Additional information**
**Correspondence and requests for materials** should be addressed to Robin J. L. Timmis or Brendan Dromey.

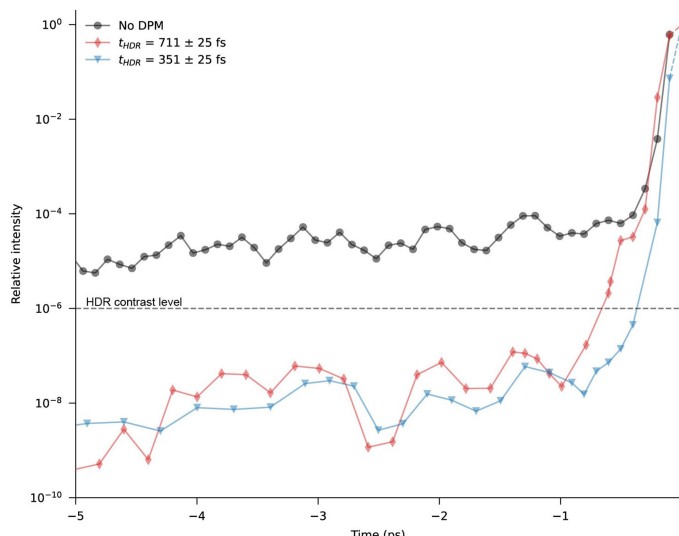

**Extended Data Fig. 1 | Long time base third-order cross correlation for confirmation of HDR contrast level selection.** As can be seen the 'fast rise time' (blue trace) and 'slow rise time' (red trace) oscillate between relative intensities of $10^{-9}$ and $10^{-7}$ in the window −5 ps to −1.5 ps. Only for timeframes closer than 1 ps to the peak do both signals show a significant increase beyond this intensity range (green circles).

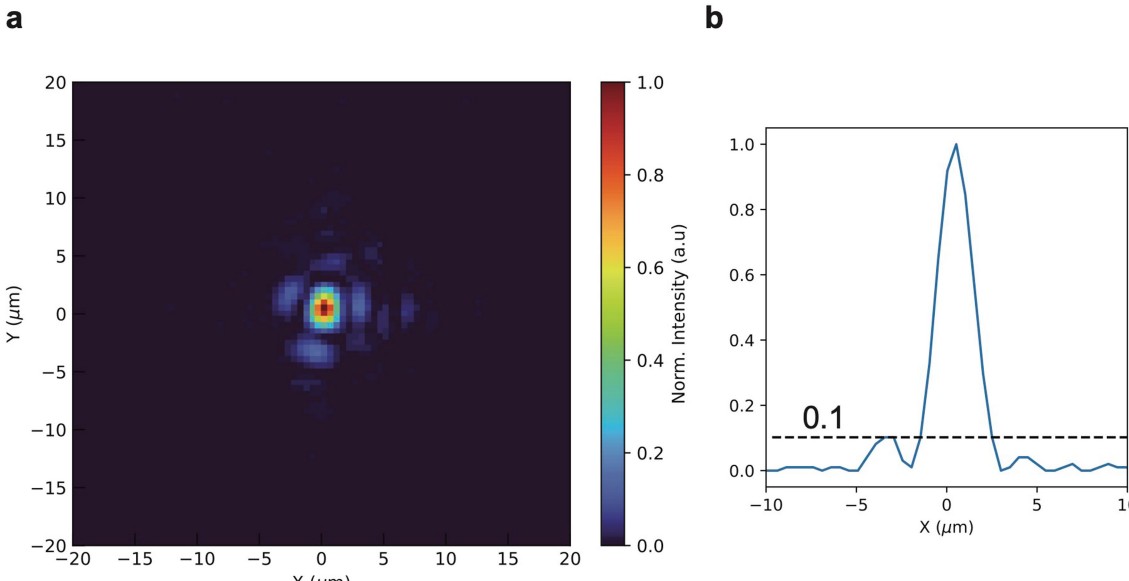

**Extended Data Fig. 2 | Gemini full beam focal spot.** (**a**) Normalised full beam spot for Gemini measuring a FWHM of ~2.3 μm. (**b**) Lineout of the focal spot showing average wing intensities up to the ~10% level.

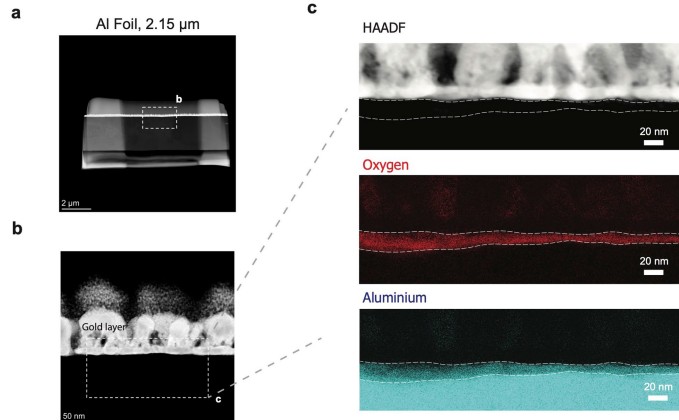

**Extended Data Fig. 3 | Aluminium oxide images.** Plot **a** shows the total lamella cut from the aluminium foil with a dashed line showing the zoomed region where **b** is imaging. Plot **b** shows a magnified area of the foil surface, which clearly shows a visible varying oxide layer on top of the black coloured bulk aluminium and below the light grey coloured protective gold layer added for the imaging process. Plot **c** shows the energy-dispersive X-ray spectroscopy (EDX) which has been used to image oxygen and aluminium. It can be seen from the red coloured region of this image that a thick region of aluminium oxide ($Al_2O_3$) has accumulated on the surface of the foils.

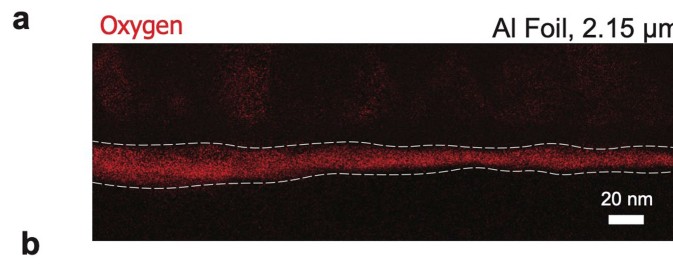

**a** Oxygen Al Foil, 2.15 μm

20 nm

**b**

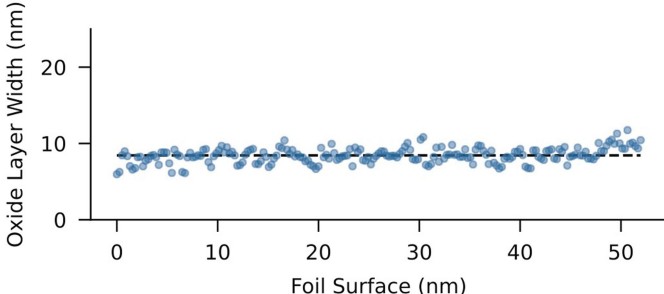

**Extended Data Fig. 4 | Thickness of aluminium oxide layer.** Plot **a** shows the oxide SEM/EDX image in Extended Data Fig. 1c. The white dashed line is a guide to the eye to show an approximate region of the $Al_2O_3$ layer. The width of the layer was calculated by measuring where the intensity falls to 1/exp(1) of its peak on either side, and the distance between them was taken as the oxide width. Plot **b** shows the spatially varying oxide thickness values across the field of view indicating an average oxide layer thickness of approximately 9 nm. However, this is just an approximation and depending on where one accepts the layer to stop (such as intensity falls to 1/2 or 1/exp(2)) and begin, this can vary from 6–16 nm.

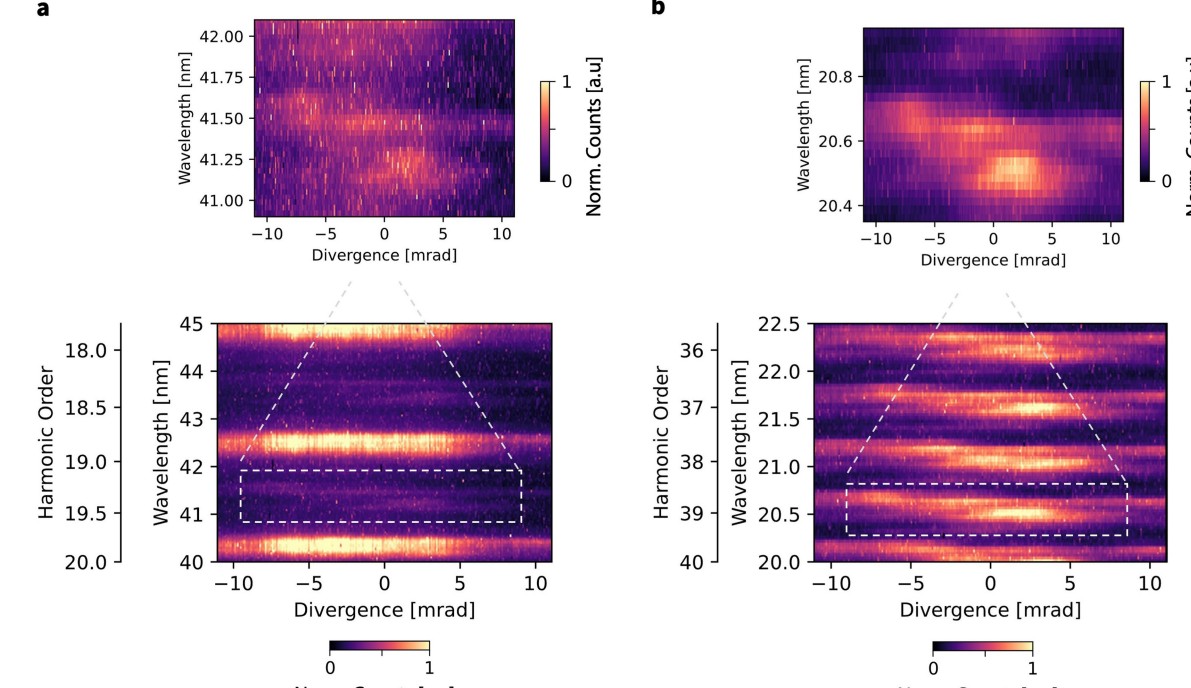

**Extended Data Fig. 5 | Higher order diffraction patterns.** Plot **a** shows the lower wavelengths which appear to have half-order harmonics. Looking at the shape of the higher order harmonics in **b** (half the wavelength range of **b**), the shape of these harmonics are almost the same as can be seen in the half order harmonics in **a**. This indicates that the half orders are in fact the higher order harmonics' second order diffraction order which have to be considered when calculating total energy in the lower orders.

**a** **b**

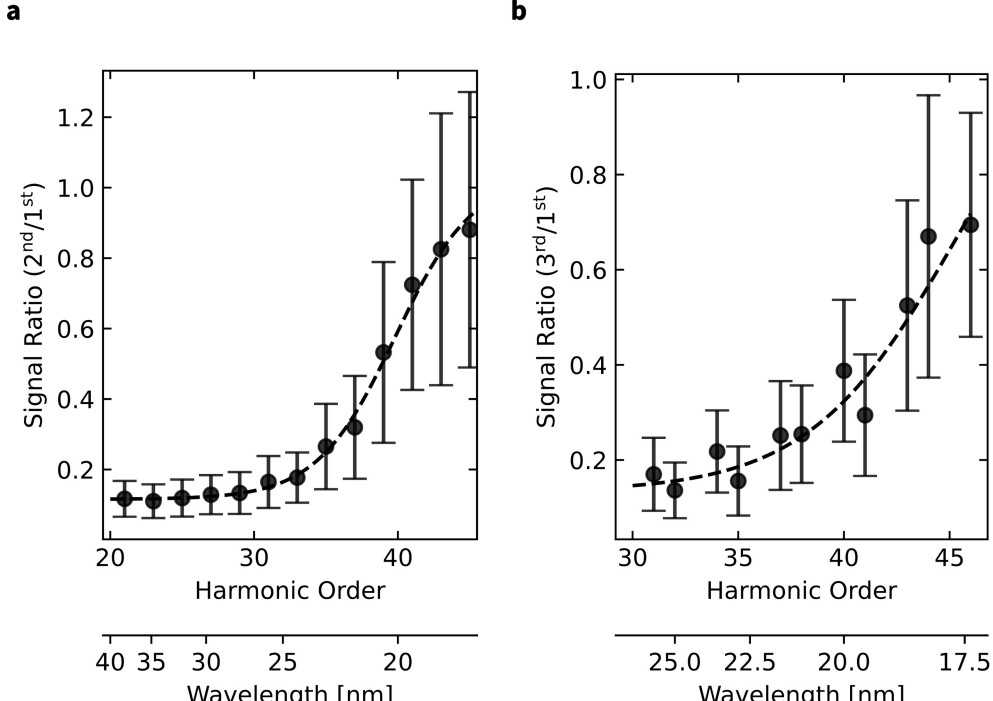

**Extended Data Fig. 6 | Second and Third Diffraction Order Efficiencies.** Plot **a** shows the ratio of the second order diffraction to the first order diffraction, from a pinhole in the aluminium filter. Plot **b** shows the ratio of third-order diffraction to first-order diffraction.

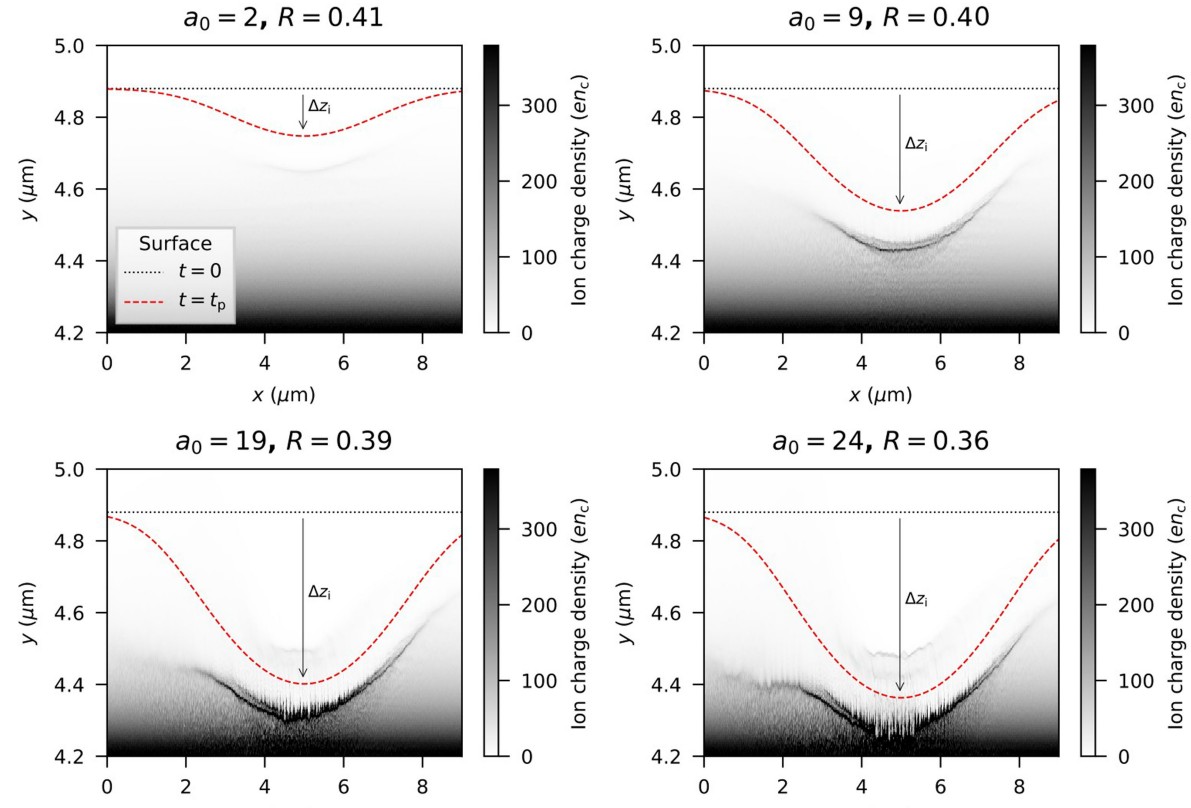

**Extended Data Fig. 7 | Numerical and analytical calculations of denting of the plasma ion surface by the laser pulse.** Simulation parameters are matched to the experimental data presented in Fig. 3. Reflectivities used in the analytical calculation are extracted from the simulations. The analytically calculated critical density surface is plotted for each $a_0$ at the start of the simulation and at the peak of the laser pulse, $t_p$.

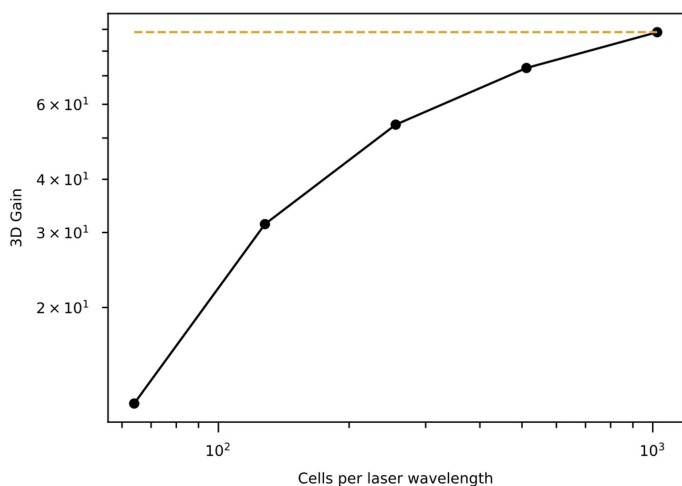

**Extended Data Fig. 8 | Convergence study of CHF gain.** It is known that high resolutions are required to accurately model the CHF. Convergence is not obtained with the highest possible simulation resolution, but the measured gain provides a lower bound on the gain that can be anticipated from the interaction.