## [Peer Review file · Nature]

Efficiency-optimised relativistic plasma harmonics for extreme fields

Corresponding Author: Dr Robin Timmis

Version 0:

Reviewer comments:

Referee #1

(Remarks to the Author)

This seminal work clarifies a decade-old riddle in relativistic high harmonic generation (RHHG). For many years it was known that when a relativistically intense laser pulse interacts with a solid target, high harmonics of very high order can be generated. However, the experimentally observed efficiencies consistently fell far below theoretical predictions and fluctuated widely from experiment to experiment, limiting the practical use of RHHG.

The present study demonstrates conclusively that this instability was due to inadequate laser pulse quality on the femtosecond timescale. By fine-tuning the temporal profile of petawatt-class laser pulses, the authors achieve excellent agreement between experiment and theory across a broad range of harmonic orders. Their results show that the interaction has now entered the efficiency saturation regime at the highest intensities. This milestone achievement decisively bridges the long-standing gap between theory and experiment, and opens the path toward the enormous intensity boosts anticipated from Coherent Harmonic Focusing (CHF). Since all surface harmonics are inherently phase-locked, their constructive interference in CHF could yield unprecedented electromagnetic field strengths, potentially approaching the QED critical field in the laboratory.

Beyond efficiency scaling, the authors also observe an intensity-dependent divergence of high harmonics, which they attribute to harmonic focusing arising from ponderomotive denting of the plasma surface. While this interpretation is plausible, it is worth noting that theory (Phys. Plasmas 14, 093104 (2007) (<https://doi.org/10.1063/1.2771519>)) also predicts that the position of the “apparent reflection point” depends on the relativistic similarity parameter $S = n/(a n_{cr})$, where n is the electron density, n_{cr} the critical density, and a the local laser amplitude. Even without surface denting, this mechanism can lead to natural focusing of reflected harmonics. The authors may wish to acknowledge this theoretical work for completeness.

Overall assessment:

This manuscript is of outstanding quality. The experiments are elegant and systematic, supported by detailed comparison with high-resolution PIC simulations. The writing is clear, the figures are compelling, and the authors situate their results well within both the historical and conceptual context of strong-field laser physics. The impact is broad: this work not only establishes a long-sought efficiency benchmark for RHHG, but also sets the stage for next-generation multi-PW facilities to access new regimes of attosecond science and strong-field QED.

Recommendation:

I strongly support publication of this manuscript in Nature in its present form. The only minor suggestion is to include the additional citation mentioned above when discussing harmonic divergence and focusing.

Referee #2

(Remarks to the Author)

The authors report on efficient high-harmonic generation (HHG) in the XUV regime from a relativistically oscillating plasma mirror driven by a PW-class laser. They claim to boost the conversion efficiencies to the long-standing theoretical predictions from plasma mirrors by reaching the optimal regime for generation, mainly by tailoring the temporal intensity profile of the driving laser pulse on a sub-ps time scale using a double-plasma mirror. The authors further suggest that this approach could be used to realize a coherent harmonic focus (CHF) that can potentially pave the way for studying strong field QED phenomena.

This experimental work, backed by 2D PIC simulations, has been detailed thoroughly and clearly, with considerable attention especially to ensuring the accuracy of the energy measured in the XUV regime, such as even quantifying the oxide layer on the Al-filter in the measurement setup.

However, the scientific novelty and significance of the conceptual advance made in this work are limited.

-The effect of plasma density gradient and near-pulse-peak contrast on the HHG from plasma mirrors has been extensively explored in the prior works (refs 9, 11, 16, and 25 cited in the manuscript).

-The key claims presented in the manuscript on the optimization of harmonic yield and approaching the efficiency limit that can be potentially suitable for a CHF are largely confirmatory and build heavily on the author's own prior publications (refs 9 and 21).

-The CHF and the corresponding intensity boost were not directly measured in this work. The discussion on the expected intensity boost (potentially reaching the Schwinger limit) due to the high conversion efficiencies and realization of a CHF relies solely on 2D PIC simulations that do not take into account spatio-temporal couplings and phase distortions. The readers would benefit from a discussion on the impact of spatio-temporal distortions on the attosecond phase locking and CHF, and the direct effect on the achievable peak intensities.

-The high-dynamic range rise time, used as t_{HDR} in the manuscript, has been defined to 6-orders of magnitude lower than the main pulse intensity and presented as a crucial threshold on sub-ps timescales for pushing the conversion efficiencies to the limit. However, the manuscript lacks a clear physical rationale or reference for choosing this threshold.

Referee #3

(Remarks to the Author)

The manuscript demonstrates experimental evidence that relativistically oscillating plasma surfaces can achieve the maximum conversion efficiencies predicted by simulations. This provides the foundation for Coherent Harmonic Focus (CHF) to experimentally reach the QED regime using available lasers and intensities. This increase in efficiency was the final missing element needed to realize CHF. Numerical simulations indicate spatiotemporal compression and a CHF intensity gain of 80.

The experimental and numerical results are very impressive and represent a milestone in relativistic laser-plasma research. The measured energies between the 12th and 47th harmonics reach record-breaking levels that are essential for CHF and significantly surpass all previously reported sources. The estimated CHF intensity gain offers a promising pathway toward a new research frontier. Moreover, the presented results firmly establish relativistic plasmas as viable future attosecond sources. The manuscript is clearly written and well structured, and it is supported by extensive experimental and numerical work, including time-resolved contrast measurements after the plasma mirror, plasma density scale length dependent signals at lower intensities, angle resolved measured and simulated spectra, and simulations of CHF under the applied experimental conditions. The figures are presented with high clarity and quality.

My main questions and comments are below.

1. The 'near-time' contrast of the Gemini laser was improved by a double plasma mirror. However, the difference between the two cases of $t_{\text{HDR}} = 711 \pm 25$ fs and $t_{\text{HDR}} = 351 \pm 25$ fs is not specified. How was the improved contrast reached? It should be shortly explained in the manuscript or in the methods.
2. Estimation of the errors seems feasible, though the number of shots is not indicated. How was the relatively large error in the XUV energy estimated (9.5 ± 4.9 mJ)? How many shots were used to determine it?
3. Lines 264-270 The HDR contrast level in Fig. 1 is only valid for the highest peak intensity and at lower intensities the line is at a different (higher) position. Which should be mentioned. Consequently, the harmonics are expected to be more intense even for $t_{\text{HDR}} = 711 \pm 25$ fs, at an intensity of 7.5×10^{18} Wcm⁻².
4. Line 319 Fig. 2 The lower harmonic limit of 25 is strange compared to the lower limit of 12 in the energy measurement.

Why is it so? Does CWE start to play a role at lower intensities?

5. Line 414 The measured focus should be inserted to support the claims about the wings.

6. Line 432 'Fig. 3d shows the intensity-dependent evolution of the harmonic beam divergence'. Laser energy increase makes the spot of emission larger so its divergence in the near field (vs intensity) smaller, while a laser spot decrease makes the divergence (vs intensity) larger. Is the decrease of the focal spot included in these simulations?

7. The authors take a rather indirect approach based on spatial properties to justify the efficiency-saturation regime, yet some of the underlying arguments are not fully convincing.

Lines 480-484, 'From the simple model for divergence described in Fig. 3, these modulations are a clear indication of interference due to harmonic emission from the wings of the focal spot. This can only be achieved in experiment if the conversion efficiency is comparable across the entire focal spot'

Interference is an extremely sensitive effect and does not require comparable efficiencies across the entire focal spot to produce pronounced modulation. Even if only about 1% of the total energy originates from the wings, the resulting modulation remains highly visible. The peak intensity is $10^{21} \text{ W cm}^{-2}$, while the wings reach only 5–10% of this value, so approximately 5×10^{19} to $10^{20} \text{ W cm}^{-2}$ (see line 474-475), which is below the efficiency saturation regime.

Consequently, the presence of strong spectrospatial modulations is not clear evidence for operation in the efficiency saturation regime.

In my opinion, the strongest argument for the efficiency saturation regime is presented in Fig. 2a, where almost the same efficiency is reached at a broad range of intensities in good agreement with simulations. I do not see how this could be achieved if the saturation regime is not reached. I miss this argument from the paper and recommend discussing it.

8. Line 485-487, 'conditions for efficient harmonic generation in the presence of curved plasma surfaces are present here across the full width of the focal spot'.

Efficient harmonic generation is not reached across the entire width of the focal spot, as the intensity at the edges is consistently lower, and at around or below $10^{20} \text{ W cm}^{-2}$ intensity it falls outside this regime. It may, however, be reached in and near the focal center.

9. Line 493, 'generation of CHF-suitable conditions' are best supported by HHG spectra with the expected power law scaling and divergence increase signaling denting and focusing. Why would the spectral modulation, which originate from harmonics generated in the wings of the focus, support it?

10. Line 505 and Fig. 4a. Q-switching is indicated, but where is mode locking?

11. Fig. 4b temporal intensity plot. What is the small second pulse after every larger attosecond peak?

12. Line 508, 'ICHF $\propto a_0^3$.' Why not ICHF $\propto a_0^5$. It seems clearer to me.

13. Line 594 'No focusing optic was used.' Extra focusing of XUV radiation onto the MCP is meant? It should be explained in more detail.

14. In Appendix A2 the harmonic energy deconvolution is described, but a lot of data are missing. Values or plots vs. wavelength of the different quantities used in Eq. A1 should be inserted.

Furthermore, distance target-filter-grating-CCD are also missing.

How was σ_x , and σ_y determined for Eq. A3? Profiles are peaked, but not exactly Gaussian. How much error does this cause? Can the measured profile be used at least in one direction? How large is typically ρ_{frac} ?

15. Lines 823-825 some info is also missing here. What are the values? What is the transmission of oxide layer? What is transmission of hydrocarbon? How is the energy uncertainty determined?

16. Fig. A2, (a) seems to have a factor of 2 change in thickness while (b) is mostly constant. Why?

17. Appendix A.2.2 line 914. The identification of 2nd and 3rd grating orders is useful and confirmation that they are small is necessary. Though, the claim "This highlights the importance of accounting for second and even third order diffraction contributions when analysing and calculating the total energy distribution in the lower order spectral region." seems to be exaggerated. This is because due to ROM scaling a 2x shorter wavelength has almost 10x weaker spectral amplitude (see also Fig. 1c) and the second diffraction order is again about 10x weaker than the first for lower harmonics (Fig A4). Thus, this false signal is in the few% level. In the spectral regime, where the ratio of 2nd to 1st order starts to increase (>30 order) the short wavelength is attenuated by the filter much more than x10, so its influence is even less. If there are a few pinholes in the filter, I expect their summed area is much smaller than the filter / beam size on the filter, which should effectively suppress them. Similar argument is valid for the third order. Therefore, the error in energy measurement from higher diffraction orders should be negligible (few%) compared to the overall energy measurement error (50%). By how much is the energy value overestimated?

18. Appendix A.2.2 line 941 "Failing to account for these effects leads to overestimating the reflected harmonic energy in the lower orders where the aluminium filter attenuates the spectrum significantly."

Which orders are meant? At starts to transmit from the 12th harmonic, there the 2nd to 1st diffraction order ratio should be

0.1, extrapolating from Fig. A4. As energy values are measured only from the 12th harmonic they should not be affected.

19. Lines 68 Abstract, 'attosecond phase-locking [6]'. Ref [6] is a seminal paper, but not a relativistic interaction. The only other paper in this regard that confirms the temporal compression for a relativistic interaction is L. Chopineau et al. Spatio-temporal characterization of attosecond pulses from plasma mirrors, Nat Phys 17 968 (2021), which should be cited here.

20. Fig. 1 The contrast measurement with the plasma mirror in Fig. 1 is very impressive from a low repetition rate DPM. Can the authors also show a normal contrast measurement of the laser, and the DPM if available, on a longer delay scale?

Further smaller remarks are below.

The caption of Fig. 2 states a conversion efficiency of $0.17 \pm 0.8\%$. The error should be 0.08% as in the text (line 264).

Line 107-108 'allows for the focusing of light to smaller foci in the diffraction limit $\sim \lambda/2n$ '.

I expect the area of the focal spot is $\sim \lambda/2n$, which is not quite clear so should be stressed.

I expect the focal-spot area to scale as $\sim \lambda/2n$, but this is not clearly stated and should be emphasized.

Line 141 'relative efficiency scaling slope of $n^{-8/3}$ consistent with theory has been observed [5, 10]'

Reference [5] confirmed only n^{-4} scaling. Though, a few papers with very short laser pulses also confirmed this scaling, such as [17] and Kormin et al. Spectral interferometry with waveform-dependent relativistic high-order harmonics from plasma surfaces, Nat Commun 9 4992 (2018).

Line 225 The methods section should be cited here as many useful parameters are described there, but their values are not specified in the main text.

Line 240 'This level is marked in Fig. 1b by the horizontal black dashed line.'

It is a black continuous line.

Line 255 'For a reduction of 114% in tHDR,'

It is only 50% reduction in time.

Lines 322-323 'Here the roll-over region is defined as where the fractional gain in harmonic efficiency as laser pulse intensity is scaled up drops below 25% and does not recover'

This is not quite clear and should be reformulated.

Line 324 and line 330 'intrinsic prepulse intensity'

It would be useful to define the term 'intrinsic prepulse'.

Line 339 I am struggling with the term 'PW-class interactions', when the energy on target is 5 J and the duration is 50 fs, which gives 100 TW or probably even lower.

Line 392 and Fig. 3 'overall harmonic beam divergence'

It seems to be half divergence angle. Is it HWHM or $H\omega^{-2}$?

Line 408, 452 and some other places in the text, 'Appendix B' or 'C'. There is no Appendix B or C.

Line 557 'The incoming laser pulse propagates in the +y-direction.'

Isn't this +x direction?

Line 755, 'The energy is calculated,'

This is not the energy, but the conversion factor from counts to energy.

Eq. A2. What is 1.96? Should this be 1.6?

Line 809-810 'High-angle annular dark-field imaging (HAADF) was performed close to the surface of the foil, which clearly shows a visible varying oxide layer on top of the aluminium bulk.'

Nothing is visible in the upper part of Fig. A1c.

Line 819-820 'it differs from the 8nm (front and back surface) measured in other studies [44].'

If 8 nm at the front as well as 8 nm at the back is meant, then it agrees well.

Fig. A3, a color bar is missing.

Line 918 'By choosing a filter that transmits only the desired harmonic range and blocks longer wavelengths, one is able to suppress unwanted second or third-order contributions.'

Correct is 'blocks shorter wavelengths'.

Overall, the manuscript presents novel and significant achievements, such as the efficient generation of harmonics from relativistic plasmas and CHF gain, and includes original, high quality experimental data and simulations of high relevance. If the authors incorporate the above suggestions and adequately address my questions, I can recommend the manuscript for publication in Nature.

Version 1:

Reviewer comments:

Referee #1

(Remarks to the Author)

I have read the revised manuscript and the authors' response to the two other referees. My previous comment has been addressed satisfactorily, and the additional discussion and citation concerning harmonic focusing provide appropriate context without changing the scope of the work.

The revised manuscript presents a clear and scientifically sound experimental demonstration of relativistic high harmonic generation operating in the efficiency saturation regime. The control of the laser pulse temporal profile on femtosecond timescales is described clearly, and the role of near-time contrast and the independently timed prepulse is well supported by the data. The comparison with PIC simulations shows good agreement over a broad range of harmonic orders and driving intensities.

A key result is the observation of near-constant harmonic conversion efficiency across a wide intensity range, consistent with numerical predictions. This provides strong evidence that the efficiency limit has been reached experimentally. The accompanying measurements of beam divergence and spatial structure are consistent with this interpretation and are adequately supported by modeling.

The manuscript resolves a long-standing discrepancy between theory and experiment in relativistic surface harmonic generation and establishes experimentally accessible conditions relevant for coherent harmonic focusing.

Recommendation:

I support publication of this manuscript in Nature in its present revised form.

Referee #2

(Remarks to the Author)

Referee #3

(Remarks to the Author)

The authors have done an excellent job revising the manuscript, which now presents the breakthrough results more clearly and convincingly. All major concerns have been addressed, with only a few minor questions remaining.

A. 16. Fig. A2, (a) seems to have a factor of 2 change in thickness while (b) is mostly constant. Why?

As this is an appendix, the dashed line is a guide to the eye, intended to show the approximate location of the oxide layer. We now clarify this in the figure caption, while the exact measurements of the oxide layer are as described in Fig. A2b).

In fact, while reading the answer of the authors I recognized that Fig. A2 a.) is plotted on a much broader range along the surface than b.), which can also explain the difference.

B. 18. Appendix A.2.2 line 941 "Failing to account for these effects leads to overestimating the reflected harmonic energy in the lower orders where the aluminium filter attenuates the spectrum significantly."

Which orders are meant? Al starts to transmit from the 12th harmonic, there the 2nd to 1st diffraction order ratio should be 0.1, extrapolating from Fig. A4. As energy values are measured only from the 12th harmonic they should not be affected. ... The power law decay means the overall energy is dominated by the lower harmonic orders so not subtracting this higher diffraction order contribution would result in a significant overestimate – for example, for the 12th order this would relate to approximately a factor of > 5 ...

This is not clearly visible in Fig. R5. The 37th harmonic in third order, which has approximately the same intensity as the 36th harmonic in third order, added to the 25th harmonic in second order, which has approximately the same intensity as the 24th harmonic in second order, do not appear almost the same to the 12th harmonic.

If this issue with the 2nd and 3rd diffraction orders are so important, what is the effect of the 4th order diffraction on the 12th harmonic energy?

If the ratio of the pinhole area to the beam area is similar in magnitude to the transmission, then the 1st diffraction order alone

gives x2 larger energy than the real energy. Was this considered?

C. Lines 322-323 'Here the roll-over region is defined as where the fractional gain in harmonic efficiency as laser pulse intensity is scaled up drops below 25% and does not recover'

This is not quite clear and should be reformulated.

Line 383: We reformulate this sentence as "Here the roll-over region is defined as the regime in which the relative increase in harmonic conversion efficiency (η) with increasing intensity (I) falls below 25% i.e. $d\eta/dI \leq 0.25$ ".

This is clearer now, except the last part i.e. $d\eta/dI \leq 0.25$. This derivative is not a dimensionless quantity. I think it is enough to say that relative increase in harmonic conversion efficiency with increasing intensity falls below 25%.

D. Line 392 and Fig. 3 'overall harmonic beam divergence'

It seems to be half divergence angle. Is it HWHM or HWe^{-2} ?

The beam divergence is given as FWHM.

This should be inserted into the text in parentheses (FWHM).

E. Line 809-810 'High-angle annular dark-field imaging (HAADF) was performed close to the surface of the foil, which clearly shows a visible varying oxide layer on top of the aluminium bulk.'

Nothing is visible in the upper part of Fig. A1c.

Fig. A1c contains the same image but with different plots showing mappings from energy dispersive x-ray (EDX) imaging. This technique allows us to distinguish between the aluminium oxide and aluminium layer. Using the oxygen mapping, we then measure the thickness of the oxide layer. Therefore, nothing is visible in the top plot of Fig. A1c; the EDX technique is necessary to observe the oxide layer.

As I understand the HAADF (upper part of Fig. A1c) does not show the oxide layer while EDX (middle and lower parts of Fig. A1c) does. Then the sentence claiming that HAADF clearly shows a visible varying oxide layer should be corrected!

After answering these last comments and questions I can recommend this excellent work for publication in Nature.

Version 2:

Reviewer comments:

Referee #3

(Remarks to the Author)

The changes made to the manuscript are satisfactory. I can recommend the paper for publication in Nature.

Reply to Referee #1:

I strongly support publication of this manuscript in Nature in its present form. The only minor suggestion is to include the additional citation mentioned above when discussing harmonic divergence and focusing.

The authors would like to thank Referee #1 for strongly supporting our manuscript for publication in Nature.

We now add the highlighted reference and include a brief discussion on this in the text. We did not include it in our original manuscript as our focus was primarily on the interpretation of our results, but we agree that it is important to reference this work for a more complete physical picture, especially for the broad readership of Nature.

Lines 464-468: Added "We note that additional contributions to harmonic focusing may arise from intensity-dependent shifts of the apparent reflection point predicted by relativistic similarity theory [35,36]."

Reply to Referee #2:

This experimental work, backed by 2D PIC simulations, has been detailed thoroughly and clearly, with considerable attention especially to ensuring the accuracy of the energy measured in the XUV regime, such as even quantifying the oxide layer on the Al-filter in the measurement.

The authors would like to thank Referee #2 for acknowledging the high level of attention we have paid to these measurements to ensure the accuracy of our results. We would also like to thank Referee #2 for highlighting some important points that improve our manuscript.

However, the scientific novelty and significance of the conceptual advance made in this work are limited.

We believe that with some simple clarifications the novelty and conceptual significance of achieving high harmonic generation from relativistically oscillating plasmas in the efficiency limit can be readily demonstrated.

-The effect of plasma density gradient and near-pulse-peak contrast on the HHG from plasma mirrors has been extensively explored in the prior works (refs 9, 11, 16, and 25)

We are not aware of any work performed in the high intensity ($> 10^{20}$ Wcm⁻²), fs-class ($\lesssim 50$ fs) regime that has demonstrated mJ-level absolute energies or efficiency saturation over a wide range of driving intensities.

Conceptually, our work advances the field by fine-tuning the rising edge of the $> 10^{20}$ Wcm⁻² interaction on sub-picosecond timeframes. We achieve this by combining both of the following methods simultaneously:

- 1) Precision control of high-contrast plasma mirror performance on sub-ps timeframes (blue vs red traces, Fig. 1b) - see response to Comment 1 from Referee #3.
- 2) The use of a well-defined pre-pulse (Fig. 2a, b, c). The resulting high-fidelity control over laser pulse contrast is key to realising the *saturation regime* for harmonic generation efficiency from relativistically oscillating plasmas.

In references [9] and [11] of the original manuscript, the pulse duration was > 500 fs (0.5 ps fullwidth half max), and while reference [9] does use a double plasma mirror, reference [11] does not. Importantly, neither use an independently timed pre-pulse to optimise the harmonic signal as, for these picosecond scale interactions, the plasma evolves significantly throughout these long pulse durations. This means that optimised plasma conditions cannot be maintained for the entire pulse window. PIC simulations (Fig. SI-3, Supplementary Information, reference [11], reproduced below in Fig. R1) show that for 500 fs interactions, the plasma evolves dramatically compared to 50 fs interactions (Fig SI-3 a) and that windowing is required to return spectral profiles similar to those observed for 50 fs (Fig SI-3 b).

Reference [16] refers to work performed at $\sim 3 \times 10^{19} \text{ Wcm}^{-2}$ which our 2-D simulations show (in Fig. 2 of our manuscript) is nearly an order of magnitude below that required to reach the efficiency saturation regime. Reference [25] refers to work that does not study HHG.

Figure SI-3 Evolution of the spectrum in transmission from a 200nm DLC foil during the rise time of a 500fs (150 cycle) full width at half maximum pulse. The simulation parameters are $a_0=20$, max density $800N_c$, 100nm preplasma ramp from vacuum to solid. Figure SI-3a) is reproduced from the main manuscript for clarity and corresponds to the full temporally integrated spectrum for the 500fs and 50fs (15 cycle) pulses. Figure SI-3b) shows the spectrum from the 500fs interaction integrated over 100fs (-60 to -30 for 150 cycle pulse centred on 0 i.e. -75 (early) to +75 (late)) during the rise time of the 500fs interaction compared with the full temporal integration from the 50fs pulse interaction. The harmonic dependent scalings and the spectrum for the 500fs are normalised to $n=28$, while the 50fs spectrum is normalised to the average of the peak of the distribution.

Fig. R1 Fig SI-3, Supplementary Information, Dromey et al. *Nature Physics* 8, 804–808 (2012).

The key claims presented in the manuscript on the optimization of harmonic yield and approaching the efficiency limit that can be potentially suitable for a CHF are largely confirmatory and build heavily on the author's own prior publications (refs 9 and 21).

Referee #2 is correct that our work confirms longstanding theoretical predictions in the field and that it provides the first clear evidence that exploiting the full potential of CHF to realise a new extreme intensity regime is now, in principle at least, possible.

Concomitantly, our work provides a paradigm shift for the field by **a)** describing a new methodology (as outlined in our reply above) and **b)** unambiguously providing the conditions required for controlling the relativistic plasma medium (Figs. 1**b** and 2**a, b, c**) at intensities $> 10^{20} \text{ Wcm}^{-2}$. Our work demonstrates how both of these elements are essential for realising harmonic generation in the efficiency limit.

While this methodology does build upon our previous work and knowledge of the field, the combination of controlling double plasma mirror performance (blue vs red traces, Fig. 1**b**) with an ultrafast, independently timed pre-pulse has not been demonstrated to date (Fig. 2**b, c**). This successful implementation provides harmonic efficiencies that are orders of magnitude higher than anything that has been observed before, demonstrating unambiguously that the efficiency limit has been realised experimentally for the first time.

-The CHF and the corresponding intensity boost were not directly measured in this work. The discussion on the expected intensity boost (potentially reaching the Schwinger limit) due to the

high conversion efficiencies and realization of a CHF relies solely on 2D PIC simulations that do not take into account spatio-temporal couplings and phase distortions. The readers would benefit from a discussion on the impact of spatio-temporal distortions on the attosecond phase locking and CHF, and the direct effect on the achievable peak intensities.

Although spatiotemporal couplings are always a potential source of limited performance, it is also clear that on the timescales involved (< 50 fs), and under conditions where the surface quality is optimised for harmonic generation in the efficiency limit, the spatiotemporal profile of the surface can be fully controlled by that of the driving laser.

In the first instance, we can consider spatiotemporal properties of the laser pulse. It is well understood in the field how to manage and control these aspects of laser pulses, and it is essential that these are optimised for CHF. Studies of the absolute degree is beyond the scope of the work here.

Another area where spatiotemporal couplings can emerge is in the quality of the plasma surface that is generated. A poor-quality surface will introduce spatial and temporal distortions to phase, limiting the ability to generate a CHF. However, this is well understood. From surface smoothing and the denting mechanism to the limits on the ability to subsequently focus emitted radiation, these are detailed in Rykovanov *et al* *New J. Phys.* **13** 023008 (2011) – reference [3] from our original manuscript.

For example, previous experimental and modelling work has shown how surface smoothing takes place on few optical cycle time frames to provided diffraction limited performance from surfaces with rms roughness $\leq \lambda/20$, where λ is the incident laser wavelength (references [3] and [5] from our original manuscript). To put this into context, harmonics extending up to the 40th order (20 nm) were emitted with diffraction limited performance (i.e. beamed) from a surface that originally had rms roughness of ~ 40 nm. This rms roughness would be 2 times the wavelength at the 40th harmonic order (20 nm). In the absence of strong smoothing and shaping by the driving laser field this level of rms roughness would lead only to wide angle scatter and no beaming, let alone diffraction limited performance, contrary to the expectation based on the Rayleigh criterion.

Where Referee #2 is correct is that if the laser pulse contrast, especially on the few ps timeframes, is not controlled then sub-optimal plasma conditions can emerge that see a tension between plasma expansion and the growing ponderomotive force of the laser. This reinforces the need to use ultrashort pulses with tightly controlled contrast on sub-ps timeframes as used in our work. It is also the reason why references [9, 11] mentioned above cannot realise the efficiency saturation regimes as the optimal conditions only persist for a short window of the 0.5 ps driving laser pulse (i.e. Fig S1-3 from reference [11] in our original manuscript).

Having achieved the efficiency limit with optimised laser-plasma coupling we did not discuss this in our original manuscript. However, we recognise this is an important point for a general readership and now highlight the implications of spatiotemporal couplings for de-optimised conditions and why we do not expect these to play a significant role in our work.

Lines 573-578: Provide clarity on spatiotemporal couplings by adding “Additionally, while spatiotemporal laser-plasma couplings can play a significant role in the quality of a CHF, in the efficiency limit, simulations show that these are dominated, and so can be controlled, by the driving field of the incident laser pulse [3,44].”

-The high-dynamic range rise time, used as t_{HDR} in the manuscript, has been defined to 6-orders of magnitude lower than the main pulse intensity and presented as a crucial threshold on sub-ps timescales for pushing the conversion efficiencies to the limit. However, the manuscript lacks a clear physical rationale or reference for choosing this threshold.

In our manuscript, t_{HDR} was carefully selected as the threshold for which significant differences in the rise time profile of the two double plasma mirror performance modes (i.e. fast and slow) are observed. Earlier in time the performance of the plasma mirror performance modes are the same (to within noise). Below we provide a longer time window that confirms this as the threshold above which significant divergence in temporal performance of double plasma mirror modes is observed.

We keep our original Fig. 1b in our manuscript as the -1.5 ps to 0 ps time frame as it is the near time contrast traces, and differences between them, that are relevant for our results. However, we

now direct the reader to a discussion on this aspect of the work in the Methods section of our revised manuscript and thank Referee #2 for raising this point.

Lines 240-241: Added “see [...] Appendix A.5 of Supplementary Information for more details.”

Lines 1240-1267: Added Fig. A8 in Appendix A.5. Reproduced below in Fig. R2.

Fig. R2 Long time base cross correlation for confirmation of HDR contrast level selection. As can be seen the “fast rise time” (blue trace) and “slow rise time” (red trace) oscillate between relative intensity $\sim 10^{-9}$ and $> 10^{-7}$ in the window -5 ps to -1.5 ps. Only for timeframes closer than 1 ps to the peak do both signals show a significant increase beyond this intensity range (green circles).

Reply to Referee #3

The authors would like to take this opportunity to thank Referee #3 for a very detailed review of our manuscript. The suggestions have improved the quality of our work and helped us to clarify our central message. We are confident that we have addressed all of the questions in our detailed responses below.

1. The ‘near-time’ contrast of the Gemini laser was improved by a double plasma mirror. However, the difference between the two cases of $t_{\text{HDR}} = 711 \pm 25$ fs and $t_{\text{HDR}} = 351 \pm 25$ fs is not specified. How was the improved contrast reached? It should be shortly explained in the manuscript or in the methods.

The improved contrast was achieved by taking into account performance of plasma mirror coatings. We quantify this improvement in performance by defining HDR contrast level as outlined in Fig. R2 in our response to Referee #2 for details on this.

Put very simply, through our measurements we determined that the anti-reflection coating on the first plasma mirror (PM) of the double plasma mirror (DPM) was breaking down too early in the rise time of the native (unaltered) Gemini pulse contrast. This resulted in the ‘slow rise time’ DPM configuration ($t_{\text{HDR}} = 711 \pm 25$ fs, red trace Fig. 1b).

By replacing the first PM in the DPM set up with an uncoated SiO_2 blank, we improved the DPM performance to the ‘fast rise time’ configuration ($t_{\text{HDR}} = 351 \pm 25$ blue trace Fig. 1b). The motivation for making this change comes from a separate branch of study on ultrafast materials science that focuses on the role played by materials that are highly structured on the nanoscale in

the lifetime of excited electrons prior to material break down [J. Kennedy *et al. Phys. Rev. Lett.* **133**, 135001 (2024), M. Coughlan *et al. New J. Phys.* **22** 103023 (2020)]. This work was carried out under the auspices of EPSRC grant Ultrafast Nanodosimetry (EP/W017245/1) that we acknowledge in our original manuscript.

In our original manuscript we did not discuss the details of this as we believed that the most important information for the general readership of Nature was the absolute profile and relative rise times of the two configurations. It is important to note that although our approach of considering PM coatings worked for this specific setup it should not be viewed as a general solution, but rather evidence that achieving absolute contrast control is more nuanced than has been assumed to date. Our concern was that explaining this would detract from the central message of the manuscript which remains that with the blue contrast trace from Fig. 1b it is possible to enter the efficiency saturation regime for $> 10^{20}$ Wcm⁻² interactions.

We direct the reader to the Methods section of our revised manuscript where we now highlight briefly how we achieved this DPM performance enhancement and include the references mentioned above.

We also now list B. Dromey as a corresponding author for contact on this aspect of the work.

Line 253: Added "(see Methods)".

Lines 627-637 (Methods): Added "Laser contrast enhancement was achieved through measurements that determined that the anti-reflection coating on the first plasma mirror (PM) of the DPM was breaking down too early in the rise time of the native (unaltered) Gemini pulse contrast. This resulted in the 'slow rise time' DPM configuration ($t_{\text{HDR}} = 711 \pm 25$ fs, red trace Fig. 1b). By replacing the first PM with an uncoated substrate, we improved the DPM performance to have a $t_{\text{HDR}} = 351 \pm 25$ blue trace Fig. 1b). The motivation for making this change comes from a separate branch of study on ultrafast materials science that focuses on the role played by materials that are highly structured on the nanoscale in the lifetime of excited electrons prior to material break down [45,46]."

2. *Estimation of the errors seems feasible, though the number of shots is not indicated. How was the relatively large error in the XUV energy estimated (9.5 ± 4.9 mJ)? How many shots were used to determine it?*

In terms of total number of shots, over 1000 shots contribute to the entire data set and, by extension, the estimation of uncertainties. We have performed stability scans that provide limits on the level of shot-to-shot fluctuation (typically 80 shots at a time) while the systematic errors are based on known levels or measurements of contaminant layers (i.e. filters etc).

The uncertainty arises due to propagation of errors, with the largest contribution to uncertainty being the angular divergence.

Lines 843-849: We now explicitly discuss the uncertainties by adding "The uncertainty in the energy deconvolution is dominated by systematic uncertainty in the thickness of surface oxide (Al₂O₃), carbon contamination layers (CH) and the beam divergence measurement (ρ_{frac}). The correction is estimated by evaluating it at two extreme limits and then taking half the difference between the resulting corrected values as a symmetric uncertainty bound."

3. *Lines 264-270 The HDR contrast level in Fig. 1 is only valid for the highest peak intensity and at lower intensities the line is at a different (higher) position. Which should be mentioned. Consequently, the harmonics are expected to be more intense even for $t_{\text{HDR}} = 711 \pm 25$ fs, at an intensity of 7.5×10^{18} Wcm⁻².*

Yes, Referee #3 has interpreted this correctly. The reason why we do not discuss this in the manuscript is that there is a lot of experimental detail associated with this measurement that effectively leads to the same overall conclusion that we currently present. Accordingly, we do not believe this would be beneficial to the broad readership of Nature.

We also would like to point out that as this lower peak intensity is not in the saturation regime, the advantage is negated. It is our plan to discuss all of the details surrounding this aspect of the work in a later publication.

In line with Referee #3's suggestion we now mention explicitly that for different peak intensities, t_{HDR} will describe different absolute intensities, and, although the relative peak to t_{HDR} is a constant, this should therefore be considered carefully in the context of a given peak intensity interaction.

Line 240: Added "see Methods section".

Lines 639-642 (Methods): Added "Note that for different peak intensities, t_{HDR} will describe different absolute intensities, while the ratio remains the same. This should therefore be considered carefully in the context of a given peak intensity interaction."

4. Line 319 Fig. 2 The lower harmonic limit of 25 is strange compared to the lower limit of 12 in the energy measurement. Why is it so? Does CWE start to play a role at lower intensities?

Yes, Referee #3 has interpreted this correctly. As we are comparing signals across multiple orders of magnitude in intensity, we wanted to have high certainty that CWE is not contributing to the observed signal for the lower intensity interactions.

Lines 381-383: Clarified by adding "This harmonic range is chosen to ensure coherent wake emission (CWE) harmonics [34] are not contributing to the observed signal for the lower intensity interactions."

5. Line 414 The measured focus should be inserted to support the claims about the wings

Lines 1181-1200: We now include Fig. A7 in our revised Appendix A.5 of Supplementary Information, reproduced below in Fig. R3.

Fig. R3 Gemini full beam focal spot. (a) Normalised full beam spot for Gemini measuring a FWHM of $\sim 2.3 \mu\text{m}$. (b) Lineout of the focal spot showing average wing intensities up to the $\sim 10\%$ level.

6. Line 432 'Fig. 3d shows the intensity-dependent evolution of the harmonic beam divergence'. Laser energy increase makes the spot of emission larger so its divergence in the near field (vs intensity) smaller, while a laser spot decrease makes the divergence (vs intensity) larger. Is the decrease of the focal spot included in these simulations?

Yes, this is included. All simulations are set up to reflect the experimental interaction to within uncertainty.

7. The authors take a rather indirect approach based on spatial properties to justify the efficiency-saturation regime, yet some of the underlying arguments are not fully convincing. Lines 480-484, 'From the simple model for divergence described in Fig. 3, these modulations are a clear indication of interference due to harmonic emission from the wings of the focal spot. This can only be achieved in experiment if the conversion efficiency is comparable across the entire focal spot'. Interference is an extremely sensitive effect and does not require

comparable efficiencies across the entire focal spot to produce pronounced modulation. Even if only about 1% of the total energy originates from the wings, the resulting modulation remains highly visible. The peak intensity is $10^{21} \text{ W cm}^{-2}$, while the wings reach only 5–10% of this value, so approximately 5×10^{19} to $10^{20} \text{ W cm}^{-2}$ (see line 474–475), which is below the efficiency saturation regime. Consequently, the presence of strong spectrospatial modulations is not clear evidence for operation in the efficiency saturation regime. In my opinion, the strongest argument for the efficiency saturation regime is presented in Fig. 2a, where almost the same efficiency is reached at a broad range of intensities in good agreement with simulations. I do not see how this could be achieved if the saturation regime is not reached. I miss this argument from the paper and recommend discussing it.

In our original manuscript our emphasis in the text was more on the fact that we had to tailor the interaction to realise the maximum efficiencies for different intensities, leaving the visual comparison in Fig. 2 as evidence for realising saturation. However, Referee #3 raises an important point that the ability to achieve the same efficiency across this intensity range is the clearest argument for entering the saturation regime. We now explicitly state this in our revised manuscript.

Lines 389–390: Added “In summary, the observation of near constant harmonic conversion efficiency over a broad range of peak driving intensities is a clear indication of achieving the efficiency saturation regime.”

See reply to point 8 for a detailed discussion on spectrospatial modulations and contributions from lower intensity regions of the focal spot.

8. Line 485–487, ‘conditions for efficient harmonic generation in the presence of curved plasma surfaces are present here across the full width of the focal spot’. Efficient harmonic generation is not reached across the entire width of the focal spot, as the intensity at the edges is consistently lower, and at around or below $10^{20} \text{ W cm}^{-2}$ intensity it falls outside this regime. It may, however, be reached in and near the focal center.

Comments 7 and 8 relate to an important detail of our work that that can be understood by examining the differences between the 1-D and 2-D physical pictures for our experiment.

In short, the spectral efficiency does not saturate until a higher peak intensity in 2-D compared to 1-D as the wings *must* contribute significantly to the observed signal.

While our rollover is measured to set in at $\sim 10^{20} \text{ W cm}^{-2}$, this is *not* the same intensity predicted for efficiency rollover in the 1-D harmonic spectrum. This can be understood by examining Figure 3 from G. D. Tsakiris *et al. New J. Phys.* **8** 19 (2006) – reproduced below for completeness in Fig. R4.

The work in Tsakiris *et al.* is performed in 1-D and as can be seen in part (b) of Figure 3 in this paper, all harmonic orders enter the saturation regime at different a_0 values. It is important to note that each of the different traces in (b) has a high pass filter applied meaning that the efficiency curve is dominated by a small group of a harmonic orders at, or close to, the high-pass filter cut-off n_F .

Here it is very clear that the $\sim 53^{\text{rd}}$ harmonic enters saturation in the 1-D picture for an intensity of $\sim 7.5 \times 10^{19} \text{ W cm}^{-2}$ ($\sim a_0 = 6$). This scales to $\sim 5 \times 10^{19} \text{ W cm}^{-2}$ for a harmonic order of ~ 45 and $< 10^{19} \text{ W cm}^{-2}$ for harmonic order 25. Crucially, the 25th harmonic is the order that will dominate our observations in Fig. 2.

Conversely, as can be seen in our Fig. 2a, the 2-D simulations predict that we only enter the saturation regime for peak intensity $\sim 10^{20} \text{ W cm}^{-2}$ for orders 25 and higher. This is approximately an order of magnitude higher intensity than that required for the 25th harmonic order to enter efficiency saturation as predicted by 1-D simulations. By the time we reach a peak driving intensity of $\sim 10^{21} \text{ W cm}^{-2}$, this is nearly 2 orders of magnitude higher than the efficiency saturation intensity predicted for the 1-D saturation regime.

Figure 3. The efficiency η_{XUV} for the conversion of the laser light into XUV attosecond pulses (a) as a function of the filter cut-off harmonic n_F for $a_L = 10$ and three values of the exponent q and (b) as a function of the normalized filed amplitude a_L for $q = 5/2$ and the indicated filter cut-off harmonic n_F . In both cases the XUV efficiency is calculated using equation (5) in combination with equation (3).

Fig. R4 Figure 3 from G. D. Tsakiris *et al. New J. Phys.* **8** 19 (2006).

In summary, to achieve efficiency saturation in 2-D simulations, strong contributions from the lower intensity regions of the focal spot are required. This means that the peak intensity required to achieve this shifts to higher intensities. This is what our simulations in Fig. 2a show. It is therefore accurate to state that strong spectrospatial modulations are a clear indication that efficiency saturation has been achieved for our experimental conditions.

In our original manuscript we used 2-D simulations specifically to include this important detail when developing our conclusions.

9. Line 493, 'generation of CHF-suitable conditions' are best supported by HHG spectra with the expected power law scaling and divergence increase signaling denting and focusing. Why would the spectral modulation, which originate from harmonics generated in the wings of the focus, support it?

As discussed above, spectrospatial modulations are evidence that we are in the saturation regime as the wings are contributing to the observed signal.

10. Line 505 and Fig. 4a. Q-switching is indicated, but where is mode locking?

Lines 513 and 538: We now include mode-locking for completeness. As Q-switching and mode-locking both sit within the envelope of initial steep increase/leap for focused intensity we had originally kept the figure simple.

11. Fig. 4b temporal intensity plot. What is the small second pulse after every larger attosecond peak?

The relativistic plasma interaction is a highly dynamic, strongly nonlinear medium for attosecond pulse train generation in which multiple mechanisms can play a role in the generation of extreme ultraviolet radiation. The small secondary pulses are at the few % level (and as such do not contribute significantly) and are likely due to emission from secondary electron nanobunches formed during macroscopic oscillation of the plasma surface. A detailed investigation of this emission is beyond the scope of the present work but will be the subject of future campaigns as we look to implement CHF at major multi-PW facilities such as ELI and Vulcan 20 PW.

12. Line 508, 'ICHF $\propto a_0^3$.' Why not ICHF $\propto a_0^5$. It seems clearer to me.

Since a_0 is dimensionless it is easier to understand for the general readership of Nature by keeping the units the same on both sides of the proportionality.

However, we see Referee #3's point that this is a potential area for confusion and so we now express this proportionality as an "intensity gain" for the Coherent Harmonic Focus.

Line 518: Rewritten as $I_{\text{CHF}} / I \propto a_0^3$.

13. Line 594 'No focusing optic was used.' Extra focusing of XUV radiation onto the MCP is meant? It should be explained in more detail.

We now clarify this statement by explaining that the XUV signal is incident directly onto our detector.

Line 616: Reworded as "No focusing optic was used so that the XUV signal is incident directly onto the CCD."

14. In Appendix A2 the harmonic energy deconvolution is described, but a lot of data are missing. Values or plots vs. wavelength of the different quantities used in Eq. A1 should be inserted. Furthermore, distance target-filter-grating-CCD are also missing. How was σ_x , and σ_y determined for Eq. A3? Profiles are peaked, but not exactly Gaussian. How much error does this cause? Can the measured profile be used at least in one direction? How large is typically ρ_{frac} ?

Although we recognise that it is important to be fully transparent wherever possible, it is also standard practice in the field to provide thickness of filters and oxide layers and the reference/link to the sources of the transmission/absorption values for these i.e. Henke tables.

We provide the model no. of our CCD in the methods, and its Quantum Efficiency is freely available online. Indeed, we are unsure if we are legally allowed to present this data as part of a publication. Certainly, the grating efficiency curves provided by Shimadzu for the grating that we use (see *Supplementary Information Appendix A.2.2* for details) are all marked confidential. Again this information is freely available from the Supplier (Laminar-type Replica Diffraction Gratings for VUV/Soft X-ray Region | SHIMADZU CORPORATION).

Line 616: Added the distance detail "placed 1.2 m from the interaction point" to the revised Appendix.

Lines 786 and 788: Added the references [54] and [55] respectively with links to the relevant sources used for energy deconvolution.

With respect to the beam being "not exactly Gaussian" we have extensive data that measures the signal at wide angles, reducing the uncertainty to $\sim 10\%$. The exact range is in the manuscript and is harmonic specific. In terms of ρ_{frac} , this again depends on the harmonic order and can be calculated from Eq. A5 in *Appendix A.2* using the measured beam divergences, which again are harmonic specific.

Line 813: To clarify how σ_x and σ_y were determined we added "measured values of the beam divergence are used".

15. Lines 823-825 some info is also missing here. What are the values? What is the transmission of oxide layer? What is transmission of hydrocarbon? How is the energy uncertainty determined?

For this information we provided references [43 and 44] in our original manuscript, (now references [53 and 56]). Again, more generally, all information is freely available through CXRO and the Henke tables [53 in our revised manuscript]. We provide thicknesses and constituents for these contaminant layers. It is impractical to reproduce this information in its entirety and is not standard practice in the field to do so. Relevant additions to the text are presented in the response to comment 2 (**lines 843-849** in the manuscript).

16. Fig. A2, (a) seems to have a factor of 2 change in thickness while (b) is mostly constant. Why?

As this is an appendix, the dashed line is a guide to the eye, intended to show the approximate location of the oxide layer. We now clarify this in the figure caption, while the exact measurements of the oxide layer are as described in Fig. A2b).

Lines 928-929: Added "The white dashed line is a guide to the eye to show an approximate region of the Al_2O_3 layer".

17. Appendix A.2.2 line 914. The identification of 2nd and 3rd grating orders is useful and confirmation that they are small is necessary. Though, the claim “This highlights the importance of accounting for second and even third order diffraction contributions when analysing and calculating the total energy distribution in the lower order spectral region.” seems to be exaggerated. This is because due to ROM scaling a 2x shorter wavelength has almost 10x weaker spectral amplitude (see also Fig. 1c) and the second diffraction order is again about 10x weaker than the first for lower harmonics (Fig A4). Thus, this false signal is in the few% level. In the spectral regime, where the ratio of 2nd to 1st order starts to increase (>30 order) the short wavelength is attenuated by the filter much more than x10, so its influence is even less. If there are a few pinholes in the filter, I expect their summed area is much smaller than the filter / beam size on the filter, which should effectively suppress them. Similar argument is valid for the third order. Therefore, the error in energy measurement from higher diffraction orders should be negligible (few%) compared to the overall energy measurement error (50%). By how much is the energy value overestimated?

See reply to comment 18 for complete response.

18. Appendix A.2.2 line 941 “Failing to account for these effects leads to overestimating the reflected harmonic energy in the lower orders where the aluminium filter attenuates the spectrum significantly.” Which orders are meant? Al starts to transmit from the 12th harmonic, there the 2nd to 1st diffraction order ratio should be 0.1, extrapolating from Fig. A4. As energy values are measured only from the 12th harmonic they should not be affected.

Referee #3 correctly notes that the power law decay and 10x weaker 2nd diffraction order results in a relative signal strength on the % level. However, the transmission of the filter, including oxide and carbon contaminants, and the camera response, drop sharply for the lowest harmonic orders. For the very lowest harmonics (12th and 13th) the relative difference compared to 24th and 26th and 36th and 39th is approximately 2 orders of magnitude resulting in a final signal level that is comparable when considering overlapping 2nd and 3rd diffraction orders. The power law decay means the overall energy is dominated by the lower harmonic orders so not subtracting this higher diffraction order contribution would result in a significant overestimate – for example, for the 12th order this would relate to approximately a factor of > 5. At higher harmonic orders, the magnitude of this effect is reduced as the relative filter transmission is closer.

Fig. R5 Typical spectrum through 3 microns of Al showing the relative strengths of 1st, 2nd and 3rd order at the specified spectral position.

As an example, in Fig. R5 above is an image of the 12th to 13th harmonic for an optimised shot through a 1.5 micron Al filter. We can see the 25th harmonic in 2nd diffraction order and the 37th and 38th in 3rd diffraction order in the region between the harmonics. The higher diffraction orders are comparable in signal level to each other and, when combined at the position of an integer harmonic, will be a significant contribution.

[Text redacted]

[Figure redacted]

[Text redacted]

19. Lines 68 Abstract, 'attosecond phase-locking [6]'. Ref [6] is a seminal paper, but not a relativistic interaction. The only other paper in this regard that confirms the temporal compression for a relativistic interaction is L. Chopineau et al. Spatio-temporal characterization of attosecond pulses from plasma mirrors, *Nat Phys* 17 968 (2021), which should be cited here.

Line 68: We now include this reference as [7].

20. Fig. 1 The contrast measurement with the plasma mirror in Fig. 1 is very impressive from a low repetition rate DPM. Can the authors also show a normal contrast measurement of the laser, and the DPM if available, on a longer delay scale?

In response to Referee #2's comment above we provide this information in Fig. R2 extending to 5 ps. We do not modify our original Fig. 1b as the difference between the red and blue traces is central to the main message of our work and clarity is crucial for anyone attempting to reproduce our results.

We now include this information as Fig. A8 in a new *Appendix* section A.5 to clarify for a more specialised readership the exact reasoning behind the choice of our HDR contrast level and the main laser profile on 5 ps time frames.

Further smaller remarks are below.

The caption of Fig. 2 states a conversion efficiency of $0.17 \pm 0.8\%$. The error should be 0.08% as in the text (line 264).

We thank Referee #3 for noticing this error.

Line 208: It has now been amended.

Line 107-108 'allows for the focusing of light to smaller foci in the diffraction limit $\sim \lambda 2n$ '. I expect the area of the focal spot is $\sim \lambda 2n$, which is not quite clear so should be stressed. I expect the focal-spot area to scale as $\sim \lambda 2n$, but this is not clearly stated and should be emphasized.

Line 107: We have now stated explicitly that we are referring to the focal spot area by modifying this to "allows for the focusing of light to a smaller focal spot area in the diffraction limit".

Line 141 'relative efficiency scaling slope of $n^{-8/3}$ consistent with theory has been observed [5, 10]' Reference [5] confirmed only n^{-4} scaling. Though, a few papers with very short laser pulses also confirmed this scaling, such as [17] and Kormin et al. Spectral interferometry with waveform-dependent relativistic high-order harmonics from plasma surfaces, Nat Commun 9 4992 (2018).

Line 141: We have changed reference [5] to be reference [12] (in our revised manuscript), and we have included Kormin et al (reference [23]).

Line 225 The methods section should be cited here as many useful parameters are described there, but their values are not specified in the main text.

We have included a reference to the methods section.

Line 225: Added "(see Methods for details)."

Line 240 'This level is marked in Fig. 1b by the horizontal black dashed line.' It is a black continuous line.

We can see how the line appears to be solid when it is in fact a dense dotted line.

Line 242: We have amended the text to reflect that it is a dotted line as such.

Line 255 'For a reduction of 114% in t_{HDR},

Line 257: We clarify this in the text as follows "for a reduction of ~ 360 fs in t_{HDR}".

Lines 322-323 'Here the roll-over region is defined as where the fractional gain in harmonic efficiency as laser pulse intensity is scaled up drops below 25% and does not recover' This is not quite clear and should be reformulated.

Line 383: We reformulate this sentence as "Here the roll-over region is defined as the regime in which the relative increase in harmonic conversion efficiency (η) with increasing intensity (I) falls below 25% i.e. $d\eta/dI \leq 0.25$ ".

Line 324 and line 330 'intrinsic prepulse intensity' It would be useful to define the term 'intrinsic prepulse'.

Line 296-298: We define the intrinsic prepulse by adding "that is, the native prepulse and temporal pedestal of the laser system prior to contrast enhancements".

Line 339 I am struggling with the term 'PW-class interactions', when the energy on target is 5 J and the duration is 50 fs, which gives 100 TW or probably even lower.

We understand this concern, however our choice of 'PW class' was intended to refer to the system rather than the interaction directly. The only way to currently achieve intensities $\sim 10^{21}$ Wcm⁻² over a double plasma mirror with losses $\sim 50\%$ over the DPM setup is to use a PW class system.

Line 306: We replace the occurrence of PW-Class interaction with "these high-intensity conditions."

Line 392 and Fig. 3 'overall harmonic beam divergence' It seems to be half divergence angle. Is it HWHM or HWe⁻²?

The beam divergence is given as FWHM.

Line 408, 452 and some other places in the text, 'Appendix B' or 'C'. There is no Appendix B or C.

Amended to be all *Appendix A*.

*Line 557 'The incoming laser pulse propagates in the +y-direction.'
Isn't this +x direction?*

We thank Referee #3 for spotting this typo.

Line 557: It has been corrected from **+y** to **+x**.

*Line 755, 'The energy is calculated,'
This is not the energy, but the conversion factor from counts to energy.*

Line 791: We amended this to read "energy per count" for Eq. A2.

Eq. A2. What is 1.96? Should this be 1.6?

Line 794: We have now replaced this typo with q_e , - Eq. A2.

Line 800: Added the definition, " q_e is the electronic charge".

Line 809-810 'High-angle annular dark-field imaging (HAADF) was performed close to the surface of the foil, which clearly shows a visible varying oxide layer on top of the aluminium bulk.'

Nothing is visible in the upper part of Fig. A1c.

Fig. A1c contains the same image but with different plots showing mappings from energy dispersive x-ray (EDX) imaging. This technique allows us to distinguish between the aluminium oxide and aluminium layer. Using the oxygen mapping, we then measure the thickness of the oxide layer. Therefore, nothing is visible in the top plot of Fig. A1c; the EDX technique is necessary to observe the oxide layer.

*Line 819-820 'it differs from the 8nm (front and back surface) measured in other studies [44].'
If 8 nm at the front as well as 8 nm at the back is meant, then it agrees well.*

Stated thickness of the oxide layers to be 7-16 nm i.e. each layer front and back totalling 14-32 nm of the Aluminium filter. Hemmers et al. [56] measured that the total thickness for both front and back oxide layers was 8 nm.

Lines 869-870: We have altered the phrasing to "however, it differs from other studies which measured the oxide layer thickness as 8 nm in total not just the front surface thickness as measured here."

Fig. A3, a color bar is missing.

This has been updated with the colormap representing the counts of the CCD.

*Line 918 'By choosing a filter that transmits only the desired harmonic range and blocks longer wavelengths, one is able to suppress unwanted second or third-order contributions.'
Correct is 'blocks shorter wavelengths'.*

Line 972: We agree with Referee #3 and have corrected the text.

Reply to Referee #1:

I support publication of this manuscript in Nature in its present revised form.

The authors would like to thank Referee #1 for acknowledging the clarity and quality of the contents of the revised manuscript and for their decision on publication.

Reply to Referee #3:

The authors have done an excellent job revising the manuscript, which now presents the breakthrough results more clearly and convincingly. All major concerns have been addressed, with only a few minor questions remaining.

The authors would like to thank Referee #3 for reviewing the changes to the manuscript and recognising the value of the presented results therein. We trust that our responses below address all minor remaining concerns.

A. 16. Fig. A2, (a) seems to have a factor of 2 change in thickness while (b) is mostly constant. Why?

As this is an appendix, the dashed line is a guide to the eye, intended to show the approximate location of the oxide layer. We now clarify this in the figure caption, while the exact measurements of the oxide layer are as described in Fig. A2b).

In fact, while reading the answer of the authors I recognized that Fig. A2 a.) is plotted on a much broader range along the surface than b.), which can also explain the difference.

Yes that is true regarding the scale of the plot which, as pointed out by the reviewer, may also explain the difference.

B. 18. Appendix A.2.2 line 941 "Failing to account for these effects leads to overestimating the reflected harmonic energy in the lower orders where the aluminium filter attenuates the spectrum significantly."

Which orders are meant? Al starts to transmit from the 12th harmonic, there the 2nd to 1st diffraction order ratio should be 0.1, extrapolating from Fig. A4. As energy values are measured only from the 12th harmonic they should not be affected.

... The power law decay means the overall energy is dominated by the lower harmonic orders so not subtracting this higher diffraction order contribution would result in a significant overestimate – for example, for the 12th order this would relate to approximately a factor of > 5 ...

This is not clearly visible in Fig. R5. The 37th harmonic in third order, which has approximately the same intensity as the 36th harmonic in third order, added to the 25th harmonic in second order, which has approximately the same intensity as the 24th harmonic in second order, do not appear almost the same to the 12th harmonic.

If this issue with the 2nd and 3rd diffraction orders are so important, what is the effect of the 4th order diffraction on the 12th harmonic energy?

If the ratio of the pinhole area to the beam area is similar in magnitude to the transmission, then the 1st diffraction order alone gives x2 larger energy than the real energy. Was this considered?

The 48th harmonic does not transmit through the Al filter as it is just beyond the L-edge so we do not see its 4th order.

Yes, this was considered when looking at the pinhole spectra, however these were only used to confirm the calibration of the grating response for multiple diffraction orders. The energy measurements were only based on shots without pinholes.

C. Lines 322-323 'Here the roll-over region is defined as where the fractional gain in harmonic efficiency as laser pulse intensity is scaled up drops below 25% and does not recover'

This is not quite clear and should be reformulated.

Line 383: We reformulate this sentence as "Here the roll-over region is defined as the regime in which the relative increase in harmonic conversion efficiency (η) with increasing intensity (I) falls below 25% i.e. $d\eta/dI \leq 0.25$ ".

This is clearer now, except the last part i.e. $d\eta/dI \leq 0.25$. This derivative is not a dimensionless quantity. I think it is enough to say that relative increase in harmonic conversion efficiency with increasing intensity falls below 25%.

We agree with this suggestion and have removed this part in the text

Lines 383: Removed 'i.e. $d\eta/dI \leq 0.25$ '.

D. Line 392 and Fig. 3 'overall harmonic beam divergence'

It seems to be half divergence angle. Is it HWHM or HWe⁻²?

The beam divergence is given as FWHM.

This should be inserted into the text in parentheses (FWHM).

This has now been added to the text.

Lines 329: Added (FWHM).

E. Line 809-810 'High-angle annular dark-field imaging (HAADF) was performed close to the surface of the foil, which clearly shows a visible varying oxide layer on top of the aluminium bulk.'

Nothing is visible in the upper part of Fig. A1c.

Fig. A1c contains the same image but with different plots showing mappings from energy dispersive x-ray (EDX) imaging. This technique allows us to distinguish between the aluminium oxide and aluminium layer. Using the oxygen mapping, we then measure the thickness of the oxide layer. Therefore, nothing is visible in the top plot of Fig. A1c; the EDX technique is necessary to observe the oxide layer.

As I understand the HAADF (upper part of Fig. A1c) does not show the oxide layer while EDX (middle and lower parts of Fig. A1c) does. Then the sentence claiming that HAADF clearly shows a visible varying oxide layer should be corrected!

We have clarified that the HAADF images the front surface of the bulk aluminium whereas it is the EDX that clearly shows the varying oxide layer.

Lines 859: Amended to: 'which images the front surface of the aluminium foil'.